JCB Journal of Cell Biology

# Arp2/3-dependent endocytosis ensures Cdc42 oscillations by removing Pak1-mediated negative feedback

Marcus A. Harrell[1]* , Ziyi Liu[2]* , Bethany F. Campbell[1] , Olivia Chinsen[1] , Tian Hong[2] , and Maitreyi Das[1]

The GTPase Cdc42 regulates polarized growth in most eukaryotes. In the bipolar yeast *Schizosaccharomyces pombe*, Cdc42 activation cycles periodically at sites of polarized growth. These periodic cycles are caused by alternating positive feedback and time-delayed negative feedback loops. At each polarized end, negative feedback is established when active Cdc42 recruits the Pak1 kinase to prevent further Cdc42 activation. It is unclear how Cdc42 activation returns to each end after Pak1-dependent negative feedback. We find that disrupting branched actin-mediated endocytosis disables Cdc42 reactivation at the cell ends. Using experimental and mathematical approaches, we show that endocytosis-dependent Pak1 removal from the cell ends allows the Cdc42 activator Scd1 to return to that end to enable reactivation of Cdc42. Moreover, we show that Pak1 elicits its own removal via activation of endocytosis. These findings provide a deeper insight into the self-organization of Cdc42 regulation and reveal previously unknown feedback with endocytosis in the establishment of cell polarity.

## Introduction

The relationship between structure and function is a core tenet of biology. This holds true for cells, which must achieve a specific shape for their function. As such, the plethora of environments and purposes that cells occupy results in a wide diversity of cellular shapes and corresponding functions. These diverse shapes are attributed to polarized growth, wherein cells organize the cytoskeleton to promote growth from specific sites (Drubin and Nelson, 1996; Glotzer and Hyman, 1995). While several works have investigated how cells polarize and acquire shape, the fundamentals of polarization are still not well understood. This is mainly because polarization involves multiple signaling proteins and pathways that are tightly regulated. Major regulators of polarized growth across most eukaryotes are the highly conserved Rho Family GTPases which consist of Cdc42, Rac, and Rho (Nobes and Hall, 1999). These GTPases are tightly regulated to promote cell motility, cell shape, and proliferation (Etienne-Manneville, 2004; Manser et al., 1994; Nobes and Hall, 1999). For example, the GTPases spatiotemporally regulate cell protrusions, speed, and direction during migration, and promote different aspects of cell differentiation in synapse formation (de Beco et al., 2018; Machacek et al., 2009; Martin-Vilchez et al., 2017). These findings indicate that the GTPases are regulated via complex higher-order pathways to ensure their proper spatiotemporal activation. Given the complexity of these

pathways and the limitations of the mammalian systems, the molecular mechanisms governing cell polarization are not well understood.

The fission yeast, *Schizosaccharomyces pombe* grows in a bipolar manner from the two cell ends. In fission yeast, growth initiates at the old end, the end that existed in the previous generation, upon completion of cell division. As the cell reaches a certain size in the G2 phase, the new end that is formed as a result of cell division also initiates growth resulting in a bipolar growth pattern (Mitchison and Nurse, 1985). Similar to most eukaryotes, Cdc42 is a major regulator of cell polarization in fission yeast (Boureux et al., 2007; Etienne-Manneville, 2004; Johnson, 1999; Miller and Johnson, 1994; Pichaud et al., 2019). Fission yeast shares the same conserved mechanisms as higher eukaryotes making it an excellent model to understand higher-order molecular pathways that promote polarization and bipolarity. During bipolar growth in fission yeast, Cdc42 activity cycles on and off in a periodic manner often resulting in oscillations (Das et al., 2012; Howell et al., 2012; Wu and Lew, 2013; Xu and Jilkine, 2018). Cdc42 oscillations are dictated by cycles of positive feedback and time-delayed negative feedback via its regulators resulting in bipolar growth (Butty et al., 2002; Das et al., 2012; Das and Verde, 2013). Cdc42 activation occurs in an anticorrelated manner between the two growing ends,

..............................................................................................................................................................................................................................

[1]Biology Department, Boston College, Chestnut Hill, MA, USA;   [2]Department of Biochemistry and Cellular & Molecular Biology, University of Tennessee, Knoxville, TN, USA.

*M.A. Harrell and Z. Liu contributed equally to this paper.   Correspondence to Maitreyi Das: maitreyi.das@bc.edu.

suggesting that the growing ends must compete for resources to sustain Cdc42 activity and growth (Chiou et al., 2018; Das et al., 2012; Das and Verde, 2013; Wu et al., 2015). Due to this competition, Cdc42 must be inactivated at one cell end for the opposite end to activate Cdc42 and vice versa (Chiou et al., 2018; Das et al., 2012). Disruption of these Cdc42 activation cycles leads to a loss of bipolar growth (Das et al., 2012; Das and Verde, 2013). It is not known how Cdc42 activity periodically returns to a cell end after each cycle of the negative feedback to allow for the oscillations.

Cdc42 positive feedback is mediated by the guanine nucleotide exchange factor (GEF) Scd1 and its scaffold protein Scd2 (Das and Verde, 2013; Lamas et al., 2020a; Wu and Lew, 2013). The scaffold Scd2 binds active Cdc42 and is thus recruited to the growing cell ends (Chang et al., 1994; Endo et al., 2003; Hercyk et al., 2019; Lamas et al., 2020a). Scd2 at the cell ends then recruits Scd1 for further Cdc42 activation thus establishing a positive feedback. Active Cdc42 can be inactivated by its GTPase activating proteins (GAPs) Rga4, Rga6, and Rga3 (Das et al., 2007; Gallo Castro and Martin, 2018; Revilla-Guarinos et al., 2016; Tatebe et al., 2008). As Cdc42 activity reaches a certain threshold it triggers time-delayed negative feedback mediated by the Pak1 kinase (Das et al., 2012; Howell et al., 2012). The Pak1 kinase (p21-activated kinase) is a serine/threonine kinase and is highly conserved in regulating growth in most eukaryotes (Manser et al., 1994; Marcus et al., 1995). The Pak1 kinase is activated and recruited to the growing cell ends when it binds active Cdc42 (Ottilie et al., 1995; Tu and Wigler, 1999). When active, Pak1 phosphorylates effector proteins to modulate a host of biological pathways (Das et al., 2012; Magliozzi and Moseley, 2021; Magliozzi et al., 2020; Molli et al., 2009; Onwubiko et al., 2023; Ottilie et al., 1995; Tu and Wigler, 1999). In the absence of Pak1 activity, Scd1 and Scd2 accumulate at the cell ends leading to increased Cdc42 activity (Das et al., 2012). In budding yeast, the Pak1 homolog Cla4 has been shown to phosphorylate Cdc24, the Scd1 homolog, to reduce its GEF activity (Gulli et al., 2000; Kuo et al., 2014; Rapali et al., 2017). Together, Cdc42 positive feedback followed by time-delayed negative feedback results in oscillations at growing cell ends.

Cdc42 regulates multiple pathways such as membrane trafficking and cytoskeletal F-actin organization (Das et al., 2009; Etienne-Manneville, 2004; Evangelista et al., 1997; Johnson, 1999; Kolluri et al., 1996; Kovar et al., 2011; Lechler et al., 2000; Nobes and Hall, 1999; Onwubiko et al., 2019, 2021; Rohatgi et al., 1999). When activated, Cdc42 in turn activates the formin, For3, to nucleate linear actin cables (Bendezú and Martin, 2011; Kovar et al., 2011; Martin et al., 2007; Burke et al., 2014). Active Cdc42 also promotes branched actin polymerization, required for endocytosis and exocytosis in yeasts, through the type 1 myosin, Myo1 (Bendezú and Martin, 2011; Gachet and Hyams, 2005; Landino et al., 2021; Lechler et al., 2000, 2001; Lee et al., 2000; Murray and Johnson, 2001; Sirotkin et al., 2005). Type 1 myosin facilitates endocytosis through membrane anchoring and force production as well as by binding the Arp2/3 complex to promote the nucleation of branched actin (Lee et al., 2000; Manenschijn et al., 2019; Pedersen et al., 2023). Genetic experimentation from our lab and

others shows that Cdc42 regulates endocytosis in fission yeast both at the division site and also at the site of cell growth, and the underlying mechanisms are still being investigated (Campbell et al., 2022; Gachet and Hyams, 2005; Hercyk and Das, 2019; Murray and Johnson, 2001; Onwubiko et al., 2019). However, it is not well understood how membrane trafficking events in turn regulate Cdc42 dynamics.

Using in vivo experiments alongside mathematical modeling, we found that Arp2/3-mediated endocytosis is required for proper removal of Pak1 to allow anticorrelated oscillations. When the branched actin nucleator Arp2/3 complex is inhibited, Pak1 stabilizes at a cell end, preventing localization of the GEF and Scd1, and disrupting the positive and negative feedback loops needed for periodic Cdc42 activity at that end. Arp2/3 complex-mediated branched actin is required for endocytosis in fission yeast. While Cdc42 has been shown to regulate actin organization, our findings provide evidence that endocytosis also directly impacts Cdc42 activity (Das et al., 2009; Murray and Johnson, 2001; Watson et al., 2014). It is widely acknowledged that proper protein localization is vital for activity. Here, we show that the timely removal of inhibition plays an equally critical role and sets the stage for positive regulators to return and continue the cycles of activity.

## Results

### The Arp2/3 complex is required for anticorrelated oscillations between growing cell ends

There are two F-actin structures in *S. pombe* during interphase: linear actin cables and branched actin networks (Kovar et al., 2011; Pelham and Chang, 2001). Actin cables transverse the length of the cell, physically connecting both ends, and are required for actin-mediated delivery (Bendezú and Martin, 2011; Martin and Chang, 2006; Martin et al., 2007; Pelham and Chang, 2001). Branched actin patches are primarily located at the growing cell ends and are required for endocytosis (Gachet and Hyams, 2005; Kovar et al., 2011). Given the role of F-actin in membrane trafficking and polarization, we asked if actin structures regulate anticorrelated Cdc42 oscillations. We used the fluorescent probe CRIB-3xGFP to specifically visualize active Cdc42 for our experiments (Tatebe et al., 2008). We quantified the competition between the ends within a cell by measuring the correlation of Cdc42 activity between the ends under all experimental conditions (Fig. 1 D). To disrupt F-actin, we treated cells with an inhibitor of actin polymerization Latrunculin A (Lat-A, Fig. 1 A). As has been reported before, we find that Cdc42 oscillations are disrupted in Lat-A treated cells, resulting in loss of CRIB-3xGFP signal at the cell ends and increased depolarized signal along the cell sides (Bendezú and Martin, 2011; Mutavchiev et al., 2016; Salat-Canela et al., 2021). Lat-A treated cells displayed an enhanced correlation of CRIB-3xGFP signal between cell ends (Fig. 1 D).

To identify the nature of actin filament involved in regulating Cdc42 oscillations, we analyzed four conditions: DMSO, Lat-A, *for3Δ*, and CK-666. DMSO treatment served as a control (Fig. 1, A and B) (Video 1). *for3Δ* mutants lack actin cables, and cells treated with the Arp2/3 inhibiting drug CK-666 lack branched

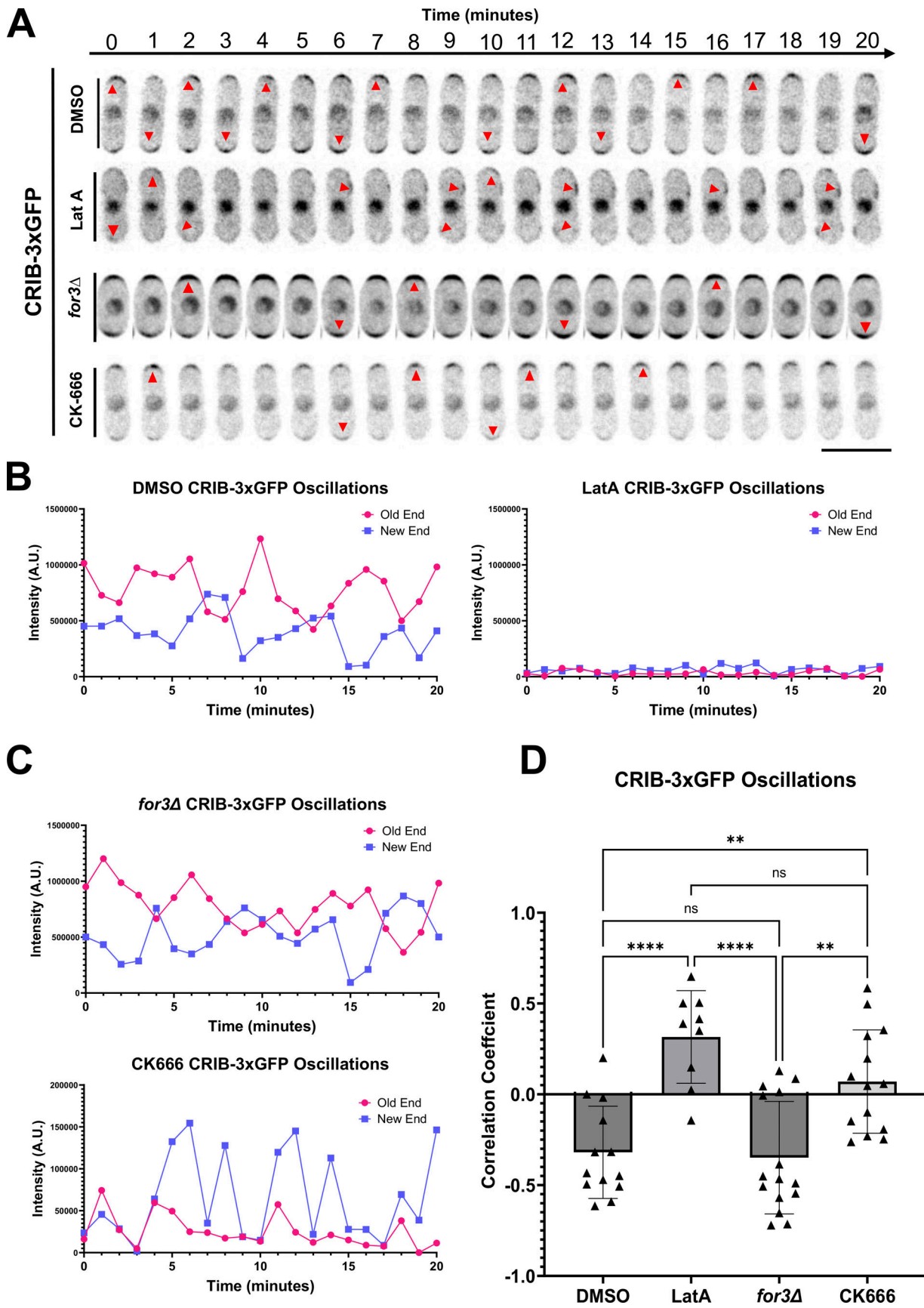

Figure 1. **Branched actin is required for competition for active Cdc42 between growing ends. (A)** Cdc42 dynamics in CRIB-3xGFP expressing cells treated with DMSO, Lat-A, CK666, and in *for3Δ* mutants. Red arrowheads mark the site of Cdc42 activation. **(B)** Representative quantification of CRIB-3xGFP

oscillations in DMSO and Lat-A-treated cells. **(C)** Representative quantification of CRIB-3xGFP oscillations in *for3Δ* and CK-666-treated cells. **(D)** Correlation coefficient of active Cdc42 oscillations between cell ends ($n \geq 9$ cells). Scale bar, 10 µm. n.s., not significant; P value, *<0.05, **<0.001, ****<0.0001, one-way ANOVA, followed by Tukey's multiple comparison test.

actin (Bendezú and Martin, 2011; Feierbach and Chang, 2001; Kovar et al., 2011; Nolen et al., 2009; Pelham and Chang, 2002; Burke et al., 2014). We then compared the level of anti-correlation between the cell ends in the absence of each actin structure (Fig. 1 D). Interestingly, we found that *for3Δ* mutants still show anti-correlated active Cdc42 oscillations, similar to DMSO-treated cells, indicating that linear actin cables do not facilitate anticorrelation between the two ends (Fig. 1, C and D). However, cells treated with CK-666, which inhibits the Arp2/3 complex and branched actin formation, do not exhibit anti-correlated oscillations (Fig. 1, C and D). Moreover, in several cells treatment with CK-666 resulted in Cdc42 activation at mostly one end (Fig. 1, A and C) (Video 2). This suggests that the Arp2/3 complex plays a role in facilitating anticorrelated Cdc42 activity between the cell ends.

Previous works have demonstrated that under stress conditions, including Lat-A treatment, Cdc42 activity is depolarized and is instead localized along the cell sides (Mutavchiev et al., 2016; Salat-Canela et al., 2021). We asked if CK-666 treatment triggered a similar stress response resulting in the loss of anti-correlation. Stress response results in the activation of the mitogen-activated protein kinase (MAP kinase), Sty1, which promotes decreased localization of Scd1 and Scd2 at the cell ends and increased localization of the Cdc42 GEF Gef1 along the cell sides results in depolarized Cdc42 activation (Hercyk et al., 2019; Mutavchiev et al., 2016). Gef1 is mostly cytoplasmic and shows increased cortical localization only in response to stress (Das et al., 2015). We analyzed CRIB-3xGFP and Gef1-mNG localization in DMSO and CK-666-treated cells. In CK-666-treated cells, we did not observe CRIB-3xGFP localization along the cell sides (Fig. S1, A and B), and Gef1-mNG continued to display cytoplasmic localization (Fig. S1, C and D). This suggests that cells must lose all F-actin structures to trigger the stress response as opposed to just branched actin. To test this, we depleted branched actin in mutants lacking linear actin cables to mimic Lat-A treatment. We treated *for3Δ* mutants with CK-666 to get rid of all F-actin structures. Indeed, we found that *for3Δ* cells when treated with CK-666 displayed depolarized CRIB-3xGFP and Gef1-mNG (Fig. S1). This suggests that CK-666 treatment by itself does not trigger a stress response.

**Inhibition of the Arp2/3 complex depletes Scd1 from cell ends**
Next, we elucidated how inhibition of the Arp2/3 complex disrupted Cdc42's regulation resulting in loss of anticorrelation between the cell ends. The Arp2/3 complex is required for branched actin-mediated endocytosis and CK-666 treatment is known to abolish endocytosis (Gachet and Hyams, 2005; Lee et al., 2000; Nolen et al., 2009; Onwubiko et al., 2019; Stamnes, 2002). Endocytosis plays a role in dispersing Cdc42 from the membrane in *S. cerevisiae* (Irazoqui et al., 2005). Endocytosis is also required to recycle proteins from the cell cortex (Gundelfinger et al., 2003; Wu et al., 2014). Inhibition of

endocytosis leads to the accumulation of these recycled proteins in the cell cortex. We asked if the Cdc42 regulators were similarly recycled by endocytosis. We visualized fluorescently labeled Cdc42 regulators in DMSO-treated and CK-666-treated cells (Fig. 2 A). Scd1-mNG (primary GEF), Scd2-GFP (Scd1 Scaffold), and Rga4-GFP (primary GAP) were imaged using fluorescent microscopy (Fig. 2 A) (Chang et al., 1994; Das et al., 2007). The Cdc42 GAP, Rga4-GFP did not show any change in its localization in CK-666 treated cells. Contrary to the expectation of endocytosis-driven recycling, we found that Scd1 and its scaffold Scd2 are significantly depleted from growing ends upon Arp2/3 complex inhibition (Fig. 2, B and C). This suggests that Scd1 and Scd2 are not recycled from the cell cortex by endocytosis, rather, endocytosis appears to enhance their localization at the cell ends. Scd1 levels are known to oscillate between the cell ends similar to active Cdc42 (Das et al., 2012). As a result, when imaging Scd1 at any given point in time, it either accumulates at any one cell end or is distributed between the two ends. Indeed, in a DMSO-treated population, snapshots of Scd1-mNG show 53% of cells with bipolar localization. In CK-666-treated cells, in addition to a decrease in levels of Scd1-mNG, we also observed bipolar localization in only 37% of cells. This suggests that in the absence of branched actin, Scd1 localization is restricted possibly due to the disruption of its oscillation.

**Cdc42-GTP inhibits the accumulation of Scd1 through an intermediary molecule**
To relate our observations on polarization regulators under perturbed conditions to the core oscillator controlling Cdc42-GTP dynamics in a rigorous manner, we constructed two models to explore potential mechanisms at the systems level. With a framework combining ordinary and partial differential equations (ODE-PDE), a previous model used an assumption of Cdc42 autocatalysis and an additional negative feedback loop between Cdc42 and Scd1 to explain Cdc42 oscillations (a classical structure chemical oscillator) (Novák and Tyson, 2008; Xu and Jilkine, 2018). However, there is no experimental evidence supporting the autocatalytic function of Cdc42 to date. We, therefore, removed this assumption and implemented a regulatory network (see Model 1, bottom of PDF) describing Scd1-mediated Cdc42 activity cycle, an experimentally supported positive regulation of Scd1 by Cdc42-GTP, and a hypothetical, direct negative feedback loop between Cdc42-GTP and Scd1 (negative feedback loop is required for oscillation) (Fig. 3 A i). In this ODE-PDE model, the two ends of the cell contain reactions at the membrane (Fig. 3 A), whereas the molecules in the cytoplasm or at the membrane on the sides are only allowed to diffuse (Xu and Jilkine, 2018). The ODE-PDE model therefore has three pools of Cdc42 and its regulators, one at each cell end and the third in the cytoplasm. We assigned fixed concentrations of Cdc42 and the GEF Scd1 and distributed them between the three pools depending on their recruitment and detachment rates (Fig. 3 B).

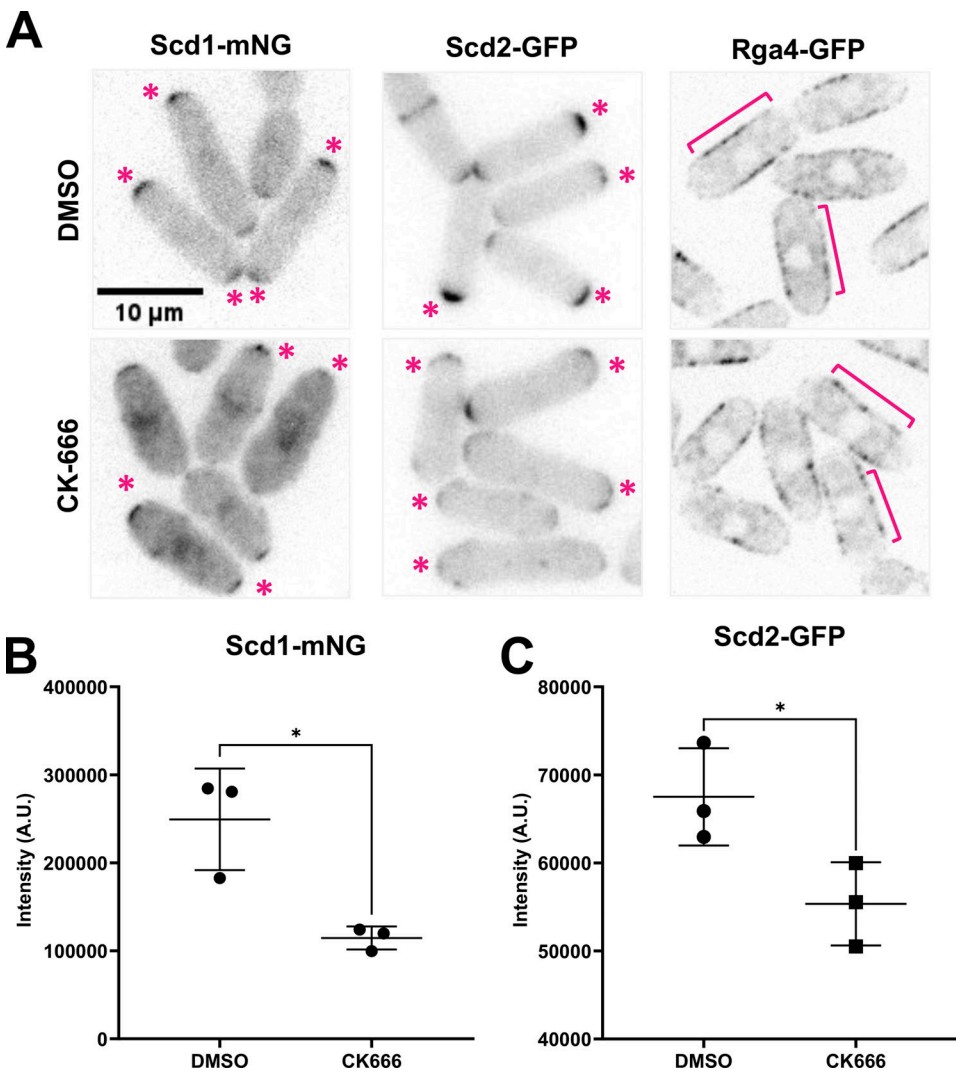

**Figure 2. Scd1 localization at the cell ends is reduced upon inhibition of the Arp2/3 complex. (A)** Localization of Scd1-mNG, Scd2-GFP, and Rga4-GFP at the ends of DMSO and CK-666-treated cells. Asterisks mark the site of Scd1-mNG and Scd2-GFP localization at the cell ends. Brackets label Rga4-GFP along the cell sides. **(B and C)** Quantification of Scd1-mNG and Scd2-GFP intensities at ends of DMSO and CK-666-treated cells ($N = 3$, $n \geq 10$ cells). Scale bar, 10 µm. n.s., not significant; P value, $*<0.05$, Student's $t$ test.

With 100,000 randomly sampled parameter values, this model failed to produce oscillatory dynamics observed experimentally (Fig. 3, A i and iii; see Model 1, bottom of PDF) (see Materials and methods). This suggests that there is another regulator that is essential for oscillation. It was known that delayed negative feedback can produce oscillations without autocatalysis (Goodwin, 1965). Furthermore, the PAK kinase has been found to inhibit Scd1/Cdc24 experimentally (Das et al., 2012; Gulli et al., 2000; Kuo et al., 2014; Rapali et al., 2017). We therefore introduced an additional unknown protein X to our model (Fig. 3 A ii; see Model 2, bottom of PDF). Protein X is activated/recruited by active Cdc42 and it inhibits the accumulation of Scd1 at the membrane, which gives rise to a delayed feedback loop. In contrast to Model 1, we found that 1.4% of randomly sampled parameter sets of Model 2 produced oscillation (Fig. 3 A iii) (see Materials and methods). Since Model 2 captured our experimental observations with respect to Cdc42 oscillatory

dynamics, we used this model and a representative oscillation-enabling parameter set to further investigate the role of branched actin in establishing anticorrelation. We were able to show oscillatory behavior with multiple parameter sets (Fig. 3 ii, Fig. S2, and Table S2). Thus, we selected the representative oscillation-enabling parameter set above arbitrarily. We assumed that the detachment rate constants ($\delta$) of each molecule positively correlate with endocytosis because branched actin in fission yeast is required for endocytosis. The scenario of branched actin disruption is described by reduced detachment rate constants in our model (i.e., decreased $\delta$) (Fig. 3 C i). To explore the influence of branched actin, we adjusted the detachment rate constants of Scd1, Cdc42, and protein X at the ends individually with the representative parameter set in Model 2 (Fig. 3 C). We found that decreasing the detachment rates of Cdc42 dampened the overall signal of Cdc42 as well as the GEF Scd1 and factor X (Fig. 3 C iii, left), but the steady state

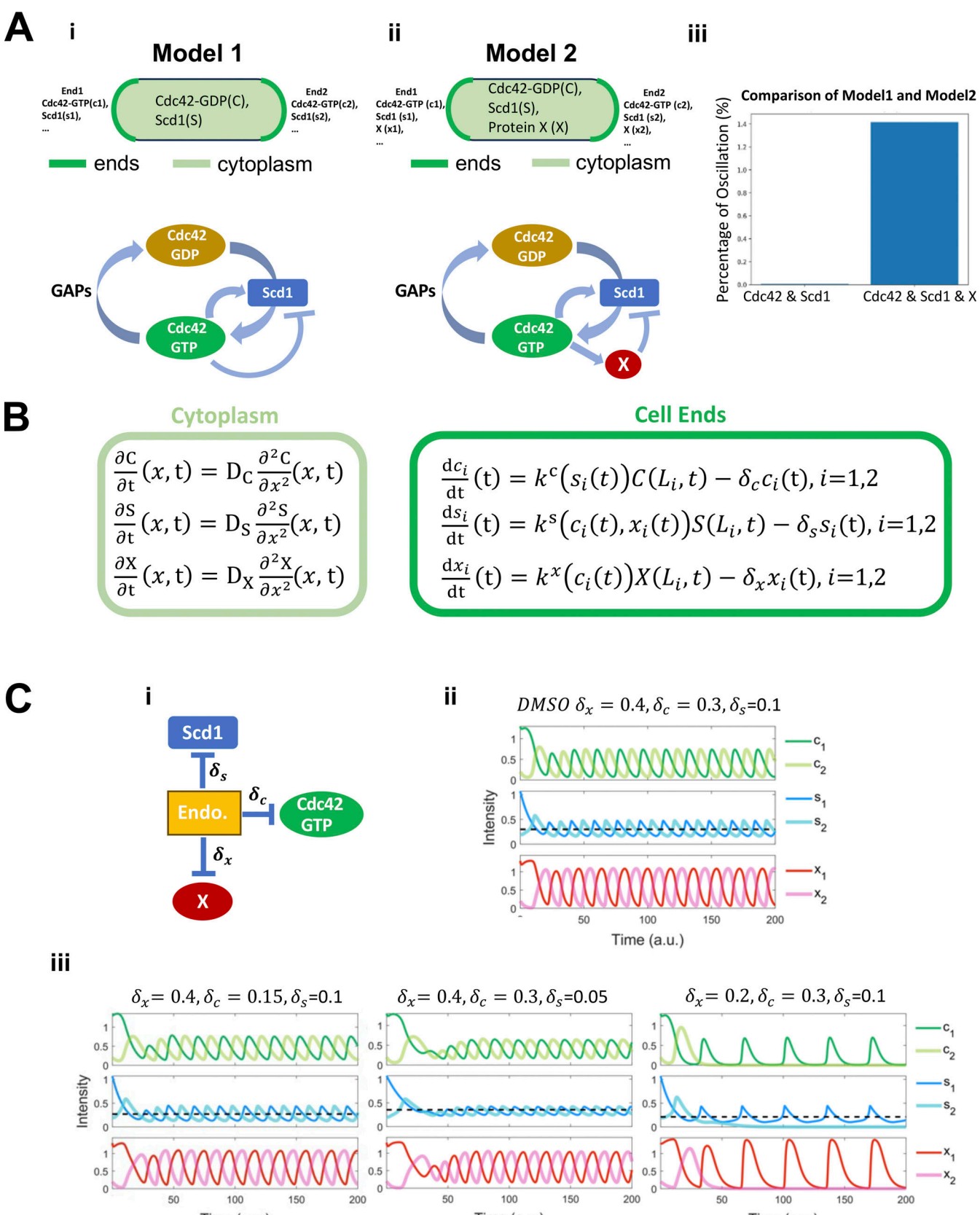

Figure 3. **Models of Cdc42 dynamics with different detachment rates. (A i)** Membranes at the cell ends accumulate Cdc42-GTP and Scd1, while the cytoplasm contains diffused Cdc42-GDP and Scd1. The panel below describes the Cdc42 activation network corresponding to Model 1. Model 1 incorporates feedback loops involving Cdc42-GTP, Cdc42-GDP, and GEF (Scd1), whereby the presence of Cdc42-GTP directly inhibits the membrane accumulation of the

GEF. **(A ii)** Membranes at the cell ends accumulate Cdc42-GTP, Scd1, and an unknown protein X, while the cytoplasm contains diffused Cdc42-GDP, Scd1 and X. The panel below describes the Cdc42 activation network corresponding to Model 2. Model 2 incorporates feedback loops involving Cdc42-GTP, Cdc42-GDP, GEF (Scd1), and X, whereby the presence of Cdc42-GTP inhibits the membrane accumulation of the GEF via protein X. **(iii)** A comparison between the percentage of oscillatory dynamics in Model 1 and Model 2. The number of parameter sets for two models is 100,000. **(B)** Reaction–diffusion equations for Model 2 (details in supplemental materials). The light green frame represents protein diffusion in the cytoplasm, and the dark green frame denotes protein binding and detachment at the cell ends. **(C i)** The effects of a drop in detachment rate on each protein in the Cdc42 activation network. **(C ii)** Numerical results of the PDE-ODE simulations for Model 2. **(C iii)** Numerical results for Model 2 were obtained by varying the detachment rates of each protein. The black dashed line is the average concentration of Scd1 at the oscillatory dynamics tip. Initial conditions at each cell end 1 and 2: $c1 = 1.3$, $c2 = 0.2$, $s1 = 1.07$, $s2 = 0.2$, $x1 = 1.3$, $x2 = 0.2$ ($c =$ Cdc42, $s =$ Scd1, $x =$ protein X).

distributions of these molecules are symmetrical at the two ends. This bipolar distribution was also observed when the Scd1 detachment rate was decreased (Fig. 3 C iii, middle). The bipolarity was observed in a wide range of detachment rate constants (Fig. S2, A and B). These simulations did not match our experimental observations of monopolar distribution upon loss of branched actin (Fig. 1). In contrast, decreasing the detachment rate constant of factor X showed monopolar dynamics of Cdc42, Scd1, and factor X itself (Fig. 3 C iii, right). Furthermore, we found that the mean level of Scd1 decreased with this perturbation, a phenomenon qualitatively reflected in our experiment (black dashed line in Fig. 3 C iii; and Fig. 2, B and C). These phenomenon with decreased detachment rate of factor x was also observed with other parameter sets (Fig. S2, C and D; and Table S2). Overall, our simulation results showed that detachment of protein X (i.e., the Scd1 inhibitor) is important for maintaining cell bipolarity and Cdc42 oscillations at both ends.

### The PAK kinase, Pak1, stabilizes when the Arp2/3 complex is inhibited

Our model predicts that decreasing the detachment of inhibitor X leads to asymmetric Cdc42 dynamics and decreased Scd1 at the cell ends. It has been reported that the Cdc42 effector PAK kinase is part of the time-delayed negative feedback and prevents GEF localization (Das et al., 2012; Das and Verde, 2013; Kuo et al., 2014). Thus, factor X could be Pak1 kinase in Model 2. Pak1 localization to the cell ends is dependent on active Cdc42 (Fig. 4, A and B). Cells with decreased active Cdc42 due to *scd1Δ* showed a decrease in Pak1-mEGFP at the cell cortex (Fig. 4 A). In *gef1Δ* mutants, Pak1-mEGFP is mostly monopolar, while in the *rga4Δrga6Δ* mutant, Pak1-mEGFP at the cell cortex is enhanced (Fig. 4 A). Similar to active Cdc42 dynamics, Pak1-mEGFP displays anticorrelated oscillations between the two cell ends (Fig. 4 D) (Video 3). We posit that when activated by Cdc42, Pak1 promotes endocytosis and this leads to its own displacement from the cell ends. Thus, we predict that in the absence of branched actin, Pak1 levels at the cell ends will stabilize and decrease in fluctuation. To test this, we used Pak1-mEGFP to visualize Pak1 localization and dynamics using timelapse microscopy (Fig. 4, C and D). We find that upon CK-666 treatment, Pak1 dynamics at the cell ends resemble that of our model 2 with a decreased detachment rate of factor X (Fig. 4 D and Fig. 3 C iii). We next measured the intensity of Pak1-mEGFP at the cell ends in DMSO- and CK-666 treated-cells. In snapshot images, we did not see a significant increase in Pak1-mEGFP levels upon CK-666 treatment, although we saw a trend in that direction (Fig. 4 E). However, when we analyzed Pak1-mEGFP dynamics at the cell

ends over time, we observed that it stabilizes with decreased fluctuations in CK-666-treated cells (Fig. 4, C, D, and F) (Video 4). To further validate whether Pak1-mEGFP showed slower dynamics upon CK-666 treatment, we performed fluorescence recovery after photobleaching (FRAP) in DMSO- and CK-666-treated cells. We bleached one half of a cell end and quantified the return of fluorescence to that end. This allows us to measure the net arrival of new Pak1-mEGFP molecules to the region. FRAP shows that Pak1-mEGFP fluorescence recovery is significantly slower in CK-666 treated cells, suggesting that Pak1-mEGFP dynamics are indeed slower upon inhibition of the Arp2/3 complex (Fig. 4 G and Fig. S5 B). Pak1-mEGFP intensity recovers at the bleached half of the cell end but not at the expense of the unbleached region next to it which suggests that recovery occurs via the arrival of cytoplasmic protein as opposed to lateral membrane diffusion.

Our model predicts that stabilizing Pak1 kinase leads to decreased Scd1 levels at the cell ends. To test this, we investigated if Scd1 intensity would be impacted by CK-666 treatment in the absence of Pak1 activity. To this end, we measured Scd1-mNG intensities in the hypomorphic *pak1* mutant *orb2-34* (*pak1-ts*) (Fig. 5 A). The *pak1-ts* mutant cells are monopolar and wider even at the permissive temperature (25°C). Moreover, in keeping with previous reports, Scd1-mNG levels at the cell end increase in *pak1-ts* mutant at permissive temperature (Fig. 5, A and B). We treated *pak1-ts* cells at permissive temperature with DMSO and CK-666 and compared Scd1-mNG intensity in these cells against *pak1+* controls under the same conditions. We found that Scd1-mNG intensity is not depleted in *pak1-ts* cells treated with CK-666 (Fig. 5, A and B). This suggests that the decrease in Scd1 levels observed in CK-666 treated cells is indeed due to Pak1 kinase activity. The *pak1-ts* cells are characteristically monopolar; however, we unexpectedly found that around 40% of these cells treated with CK-666 showed bipolar localization of Scd1-mNG (Fig. 5 A, asterisks). The mechanism of this bipolar phenotype in the absence of Pak1 is not yet understood with our existing model: in Model 2, Pak1 (X in Fig. 3) mediated negative feedback is essential for bipolarity (Fig. 5 C). We, therefore, included an additional, hypothetical negative feedback loop involving Cdc42 in our model (Model 3) (Fig. 5 D i and ii) (see details in supplemental text at bottom of PDF). Fig. 5 D ii shows the simulation results of Model 3 under both standard and decreased detachment to replicate DMSO and CK666 conditions. We next ran simulations of Model 3 without Pak1 (P = 0) (Fig. 5 E). As expected, the modeling shows monopolar dynamics in the absence of Pak1 (Fig. 5 E). However, the simulations captured bipolar phenotype when detachment rates of Cdc42 and Scd1

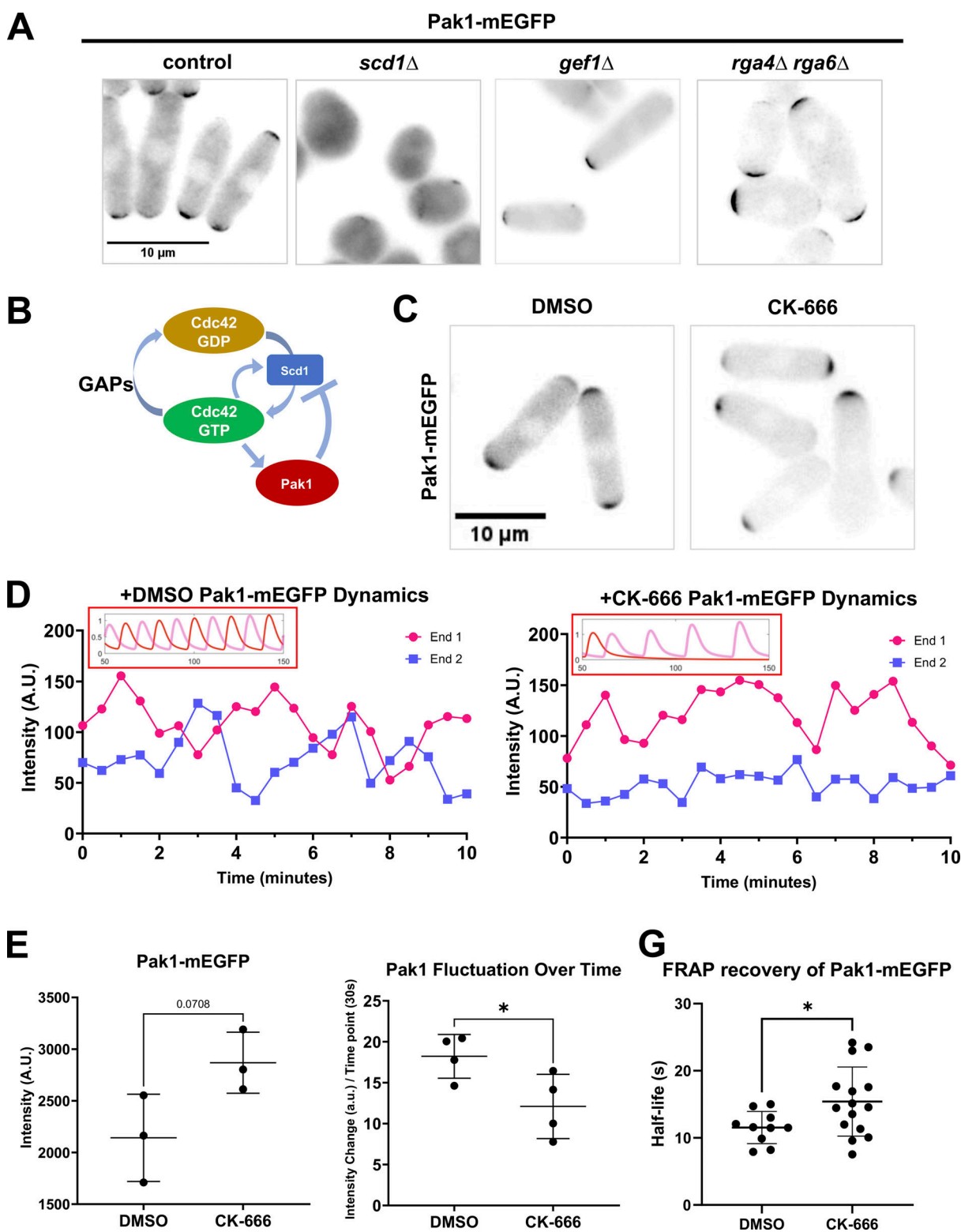

Figure 4. **The Pak1 kinase, a potent inhibitor of Scd1, is more stable at cell ends when branched actin is lost. (A)** Pak1-mEGFP localization in control cells, *scd1Δ*, *gef1Δ*, and *rga4Δ rga6Δ* mutant cells. **(B)** Model describing Cdc42 activation network where Pak1 kinase upon activation by Cdc42 triggers Scd1 detachment. **(C)** Pak1-mEGFP localization in DMSO and CK-666 treated cells. **(D)** Pak1-mEGFP dynamics in cells treated with DMSO or CK-666. Dynamics observed in vivo resembled the dynamics seen in model predictions (insets). **(E)** Quantification of Pak1-mEGFP localization at the brightest cell end in DMSO or CK-666 treated cells (*N* = 3, *n* ≥ 10 cells). **(F)** Quantification of the extent of Pak1-mEGFP fluctuations at the cell ends (*N* = 4, *n* ≥ 5 cells). **(G)** FRAP analysis showing the half-life of Pak1-mEGFP recovery in DMSO and CK-666 treated cells (*n* ≥ 10 cells). Scale bar, 10 µm. P value, *<0.05, Student's *t* test.

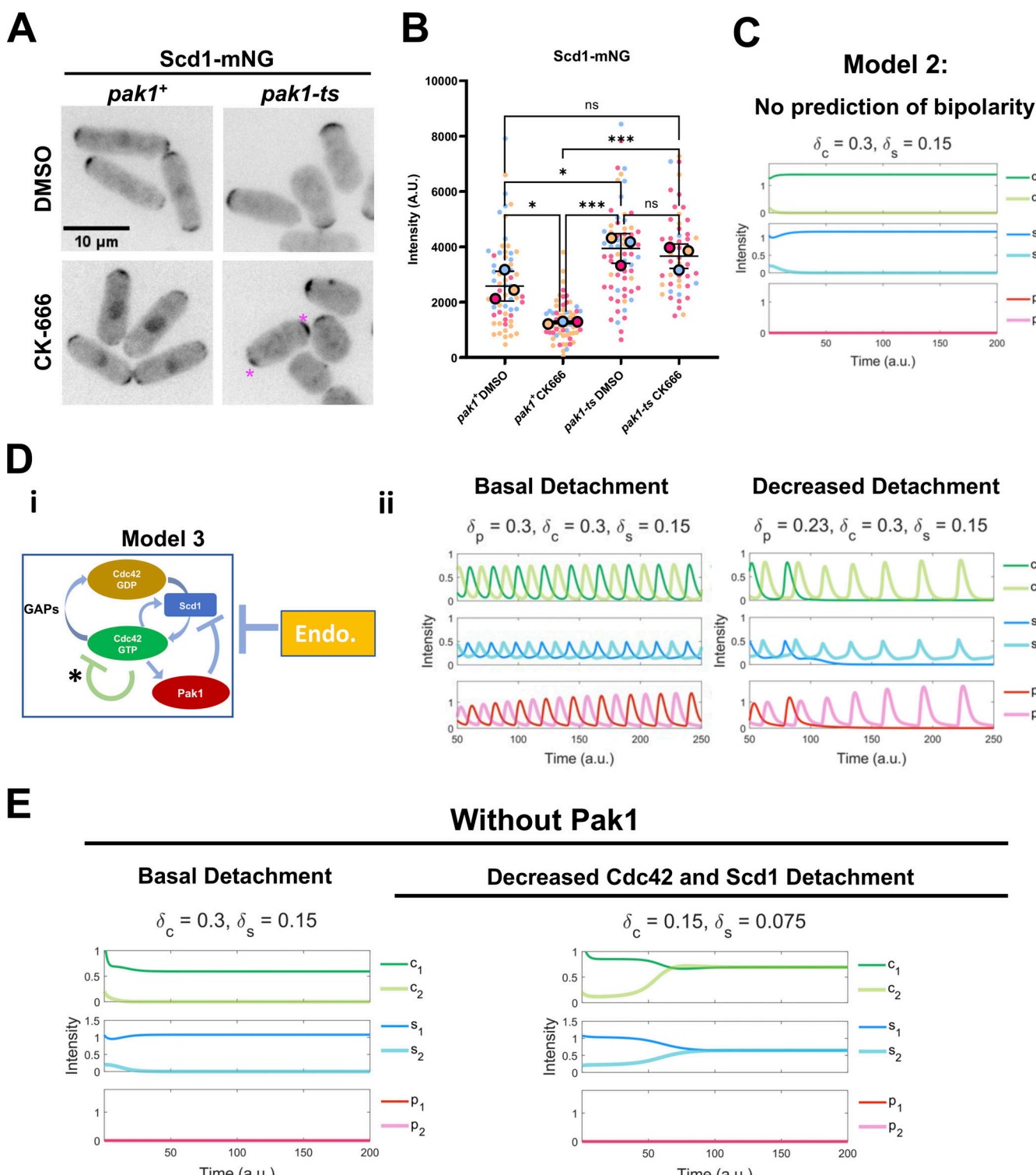

Figure 5. **Loss of Scd1 localization in CK-666-treated cells is mediated by Pak1. (A)** Scd1-mNG localization in *pak1+* and *pak1-ts* mutant cells. Asterisks depict bipolar Scd1-mNG localization. **(B)** Quantification of Scd1-mNG intensities at the cell ends in *pak1+* and *pak1-ts* cells treated with DMSO and CK-666. Colors represent each experimental replicate. Large circles are the means of each experimental replicate (N = 3, n ≥ 12 cells). **(C)** Modeling simulation of Scd1 and Cdc42-GTP localization in *pak1-ts* cells in Model 2 shows only monopolar localization of Scd1 and Cdc42-GTP in the absence of *pak1*. **(D i)** Model 3 incorporates feedback loops involving Cdc42-GTP, Cdc42-GDP, GEF (Scd1), Pak1, and endocytosis, and an additional negative signaling pathway inhibiting Cdc42-GTP. The asterisk highlights the additional new arrow to Model 3. **(D ii)** The left panel shows Model 3 simulations and the right panel shows simulations with a decreased detachment rate of Pak1. **(E)** Model 3 simulations with an additional negative signaling pathway simulated via decreased detachment rate of Cdc42 and Scd1. Scale bar, 10 μm. n.s., not significant, P value, *<0.05, ***<0.0003, one-way ANOVA, followed by Tukey's multiple comparison test.

were reduced, suggesting that bipolar and monopolar pheno-types can coexist, and a heterogeneous response to CK-666 occurred in the absence of Pak1 (Fig. 5 E).

## Pak1 accumulation at the cell ends lags behind Cdc42 activation

Oscillatory patterning requires positive feedback and time-delayed negative feedback (Cao et al., 2016; Goodwin, 1965; Howell et al., 2012; Novák and Tyson, 2008; Tsai et al., 2008; Turing, 1990). Cdc42 positive feedback comprises Scd1 and Scd2 while negative feedback is mediated by the Pak1 kinase (Chang et al., 1994; Das et al., 2012; Endo et al., 2003; Hercyk et al., 2019; Howell et al., 2012; Lamas et al., 2020a, 2020b; Ottilie et al., 1995; Wheatley and Rittinger, 2005). To this end, our model predicts that there is a spatiotemporal delay (phase shift) between the recruitment and peak accumulation of Cdc42 positive feedback proteins and the negative feedback protein, Pak1 at the growing cell ends (Fig. 6 A). Indeed, advancements in live cell microscopy have allowed for rapid timelapse imaging of even sparse proteins such as Scd1 in vivo.

To test this, we used strains with a combination of fluorescently tagged CRIB-3xGFP or CRIB-mCherry, Scd1-mNG or Scd1-tdTomato, Scd2-mCherry, and Pak1-mEGFP (Fig. 6, B and C). We found that there is practically no delay (average of 2 s) between positive feedback proteins such as CRIB-3xGFP and Scd1-tdTomato, and CRIB-3xGFP and Scd2-mCherry (Fig. 6, B and D; and Fig. S3 A). However, there was an average delay of 12 s between the accumulation of positive feedback proteins (CRIB-mCherry, Scd1-tdTomato, or Scd2-mCherry) and the peak accumulation of Pak1-mEGFP (Fig. 6, C and D; and Fig. S3 B). We validated these results in silico (Fig. 6 E). The phase shift diagrams were generated using the Hilbert Transform of the normalized data of proteins at the ends. These analyses show both raw traces and transformed traces of the protein levels. The latter highlights the oscillatory cycle of pairs of proteins and assesses for any existing phase shifts (Fig. 6 E) (Sabherwal et al., 2021). As we expected, the GEF Scd1-tdTomato peaks are in phase with those of active Cdc42 labeled by CRIB-3xGFP. Further, Pak1-mEGFP accumulation had a phase shift with respect to Scd1-tdTomato accumulation.

Current reports suggest that Pak1 kinase localization to the cell cortex is entirely dependent on active Cdc42 (Ottilie et al., 1995; Tu and Wigler, 1999). This would suggest that Pak1 kinase dynamics would be similar to that of Cdc42 activation. However, we found that Pak1 kinase shows delayed accumulation compared with active Cdc42, Scd1, and Scd2. This suggested that Pak1 dynamics are slower than that of Cdc42 activation. We tested this idea by modifying the model and experimentally measuring Pak1 dynamics. To verify if Pak1 accumulation was indeed slower than that of Cdc42, we increased the accumulation and detachment rates of Pak1 in our models. We found that with increasing rates, the oscillations dampen and phase shifts reduce to a minimal value but do not completely disappear. Furthermore, with decreasing phase shift the oscillations are ultimately extinguished (Fig. S4). The observed phase shift, or delay, between the peak accumulations of Cdc42 and Pak1 at cell ends is primarily attributed to the inherent time required for

cellular processes. This encompasses the time needed for their diffusion, subsequent accumulation, and detachment.

Next, we compared the FRAP recovery dynamics of Pak1-mEGFP with that of Scd1-mNG and Scd2-GFP. The transient binding of the CRIB-3xGFP reporter makes it unsuitable for FRAP analysis. Furthermore, measuring the dynamics of total Cdc42 will not distinguish between active and non-active Cdc42. Our data (Fig. 6 D) show that the accumulation of active Cdc42, as indicated by CRIB-3xGFP, is similar to that of Scd1-mNG and Scd2-GFP. Thus, we used Scd1-mNG and Scd2-GFP as proxies for Cdc42 activity and compared their dynamics with Pak1-mEGFP. We found that that Scd1-mNG and Scd2-GFP fluorescence recovers at the bleached region without diminishing the unbleached region next to it, similar to our observations with Pak1-mEGFP. This suggests new fluorescent molecules of Scd1, Scd2, and Pak1 replenish the bleached region via cytoplasmic exchange and not via lateral diffusion (Fig. S3 C). We found that the half-life of Pak1-mEGFP fluorescence recovery is significantly slower than the fluorescence recovery of Scd1-mNG and Scd2-GFP (Fig. 6 F and Fig. S3 C). This indicates that the dynamics and net recruitment of Pak1 kinase are slower than active Cdc42, Scd1, and Scd2.

## Loss of the Arp2/3 regulator myo1 disrupts Cdc42 oscillatory dynamics

Next, we validated if the Cdc42 oscillation defects in CK-666-treated cells are due to loss of endocytosis. Endocytosis is an essential cellular process and most endocytic mutants show severe cell growth and polarity defects thus complicating such an analysis. The Arp2/3 complex regulator myo1 is not essential and loss of myo1 results in shape and growth defects but these cells are still able to polarize (Lee et al., 2000). Thus, we used the myo1Δ mutant to investigate Cdc42 oscillatory dynamics. We found that myo1Δ mutants show similar defects in Cdc42 oscillatory dynamics to those observed in CK-666-treated cells (Fig. 7, A and B). We observed Cdc42 oscillations in myo1+ and myo1Δ mutants expressing CRIB-3xGFP. Normally, cells show anti-correlated CRIB-3xGFP oscillations with a correlation coefficient of about −0.5 (Fig. 7 B). In myo1Δ cells, CRIB-3xGFP signals at the cell ends continued to fluctuate but displayed an average correlation coefficient of about −0.2, indicating a decrease in anti-correlation (Fig. 7 B and Fig. S5 C). Next, we compared the localization of Scd1-mNG at the cell ends in myo1+ and myo1Δ cells. Similar to CK-666 treated cells, myo1Δ mutants showed a decrease in Scd1-mNG levels at the cell ends (Fig. 7, C and D). We also observed increased monopolar localization of Scd1-mNG at the cell ends in myo1Δ mutants. Our results with CK-666-treated cells suggest that the loss of Scd1 levels at the cell ends is due to the stabilization of Pak1 kinase at those ends. Thus, we analyzed Pak1-mEGFP levels and dynamics at the cell ends in myo1Δ mutants. We found that in myo1Δ mutants Pak1-mEGFP levels at the cell ends increase (Fig. 7, E and F). We observed Pak1-mEGFP dynamics using timelapse imaging in myo1+ and myo1Δ cells. Pak1-mEGFP localization mostly appeared at one cell end and displayed decreased fluctuations over time in myo1Δ cells (Fig. 7, G and H; and Fig. S5 B). This suggests that similar to CK-666-treated cells, Pak1 dynamics at the cell ends stabilized in myo1Δ

Figure 6.   **Pak1 dynamics at the cells ends are slower than that of Cdc42, Scd1 and Scd2. (A)** Simulations of our models show a delay or phase shift between the peak accumulations of Cdc42-GTP and Pak1 at the growing end. **(B)** Pairs of positive-feedback polarity proteins (CRIB-3xGFP and Scd1-tdTomato, CRIB-3xGFP and Scd2-mCherry) were simultaneously observed at the cell end. **(C)** Each positive-feedback protein (Scd1-tdTomato, CRIB-mCherry, and Scd2-

mCherry) was observed independently with the negative-feedback protein, Pak1-mEGFP. **(D)** The shift in time required to get the best correlation coefficient between signals from each pair of proteins was quantified using cross-correlation analysis (*n* ≥ 29 cell ends). **(E)** The Hilbert transform was applied to the smoothened traces of Cdc42-GTP and Scd1 for their phase reconstruction and phase shift (top). The Hilbert transform was applied to the smoothened traces of Pak1 and Scd1 for their phase reconstruction and phase shift (bottom). The thinnest traces represent normalized raw data, the thickest traces represent smoothened traces based on the normalized data, and the darkest traces represent the phase progression achieved through the Hilbert transform. **(F)** FRAP analysis shows the half-life of recovery for Scd1-mNG, Scd2-GFP, and Pak1-mEGFP (*n* ≥ 10 cells). n.s., not significant; P value, *<0.05, **<0.005, ****<0.0001, one-way ANOVA followed by Tukey's multiple comparison test.

cells. Together, these data suggest that efficient endocytosis is required for proper Cdc42 activity between the cell ends.

**Endocytic events deplete Pak1 from the membrane**
Our data suggest that the Arp2/3 complex is required for Pak1 removal to allow Scd1 localization and anticorrelated oscillations for Cdc42 between the cell ends. As a result, we hypothesize that Pak1 is depleted from the cell end via endocytosis. Pak1 kinase has been proposed to phosphorylate Myo1 for endocytosis (Attanapola et al., 2009). Thus, we posit that Pak1 kinase molecules overlapping with the endocytic patches are removed via endocytosis. To test this, we first investigated the spatial organization of Pak1 at the cell ends. Using deconvolution, we show that Pak1-mEGFP is not evenly distributed as a cap rather, Pak1 localizes as distinct puncta at the cell ends (Fig. 8 A). Next, we asked if these puncta overlapped with sites of endocytosis. Endocytic patches in fission yeast can be detected using the actin crosslinking protein Fimbrin, Fim1 (Nakano et al., 2001). Fim1 specifically localizes to branched actin patches and internalizes into the cytoplasm with the endocytic patches during endocytosis (Nakano et al., 2001). We found that the Pak1-mEGFP puncta colocalizes with Fim1-mCherry labeled endocytic patches at the plasma membrane (Fig. 8 A, left panel). Next, we analyzed Fim1-mCherry internalization in cells coexpressing Pak1-mEGFP. We observed that while the Fim1-mCherry labeled endocytic patch internalizes into the cytoplasm, Pak1-mEGFP does not (Fig. 8 A, right panel). However, we found that Pak1-mEGFP puncta overlapping with Fim1-mCherry is lost from the plasma membrane when the endocytic patch internalizes (Fig. 8 A, middle and right panels). To further verify Pak1 localization at the plasma membrane, we used Airyscan super-resolution microscopy. We observe that concentrated Pak1-mEGFP puncta overlap with Fim1-mCherry patches at the plasma membrane (Fig. 8 B, arrowheads). Current technical limitations prevent us from capturing rapid timelapse information with super-resolution Airyscan microscopy, thus we returned to spinning disk confocal microscopy for subsequent experiments. Most endocytic proteins are known to internalize with the patch. However, several regulators of endocytosis do not internalize but are simply lost from the membrane when the patch internalizes. In fission yeast, the endocytic proteins such as the F-BAR containing Cdc15 and Myo1 are lost from the membrane upon patch internalization (Arasada and Pollard, 2011; Macquarrie et al., 2019). In mutants with a delay in patch internalization, the loss of these proteins from the plasma membrane is also delayed (Onwubiko et al., 2019). It is possible that local puncta of Pak1 shows a similar loss at the membrane during patch internalization. To test if endocytic patch internalization results in the loss of Pak1 from the cell ends, we

measured the intensity of Fim1-mCherry and Pak1-mEGFP simultaneously at the plasma membrane (Fig. 8 C, dashed box). We observed that each time Fim1-mCherry starts to internalize (Fig. 8 D, dashed lines), Pak1-mEGFP decreases at the plasma membrane as indicated by reduced Pak1-mEGFP intensity at the membrane (Fig. 8, C and D). Next, we quantified when Pak1-mEGFP was lost from the membrane compared with Fim1-mCherry internalization. We found that, on average, Pak1-mEGFP is lost from the membrane within 1 s of Fim1-mCherry internalization (Fig. 8, D and E). Together, these data suggest that Pak1 is removed from cell ends when the endocytic patch internalizes.

**Pak1 activity promotes the successful internalization of endocytic patches**
While our data show that endocytosis promotes Pak1 removal from the cell ends, previous reports suggest that Pak1 regulates endocytosis in fission yeast (Murray and Johnson, 2001). To further confirm this, we analyzed endocytic dynamics in the absence of Pak1 kinase. The *pak1-ts* mutant, *orb2-34*, shows polarity defects even at its permissive temperature of 25°C. We found that successful endocytic events are significantly reduced in *pak1-ts* mutants (Fig. 9, B and E) at 25°C. Fim1-mEGFP was used to specifically visualize endocytic patches. We observed that endocytic patches still form at the membrane in *pak1+* and *pak1-ts* cells, and their lifetime at the cell membrane remains the same (Fig. 9, B and C). Thus, Pak1 does not influence the ability of endocytic patches to form. Next, we asked if endocytic patches internalize properly without Pak1 activity. Endocytic patches that internalize >350 nm away from the membrane are considered to have undergone successful scission from the plasma membrane (Basu et al., 2014). We found that there is a significant decrease in the overall distance that patches internalize in *pak1-ts* mutants compared with *pak1+* cells (Fig. 9, B and D). Furthermore, a higher fraction of the patches in *pak1-ts* mutants did not internalize beyond 350 nm, indicating a greater fraction of failed endocytic events compared with *pak1+* cells (Fig. 9, B and E). In addition, patches that do not properly internalize show three types of failed endocytic events such as incomplete internalizations, patch retractions, or stalled events (Fig. 9 B).

## Discussion
While Cdc42 activation spatiotemporally regulates actin organization for membrane trafficking and polarized growth, it is not well understood how Cdc42 itself is dynamically regulated. In fission yeast, Cdc42 and its regulators undergo oscillatory dynamics between the two cell ends, and this lends itself to the dynamic regulation of actin organization (Coll et al., 2003; Das

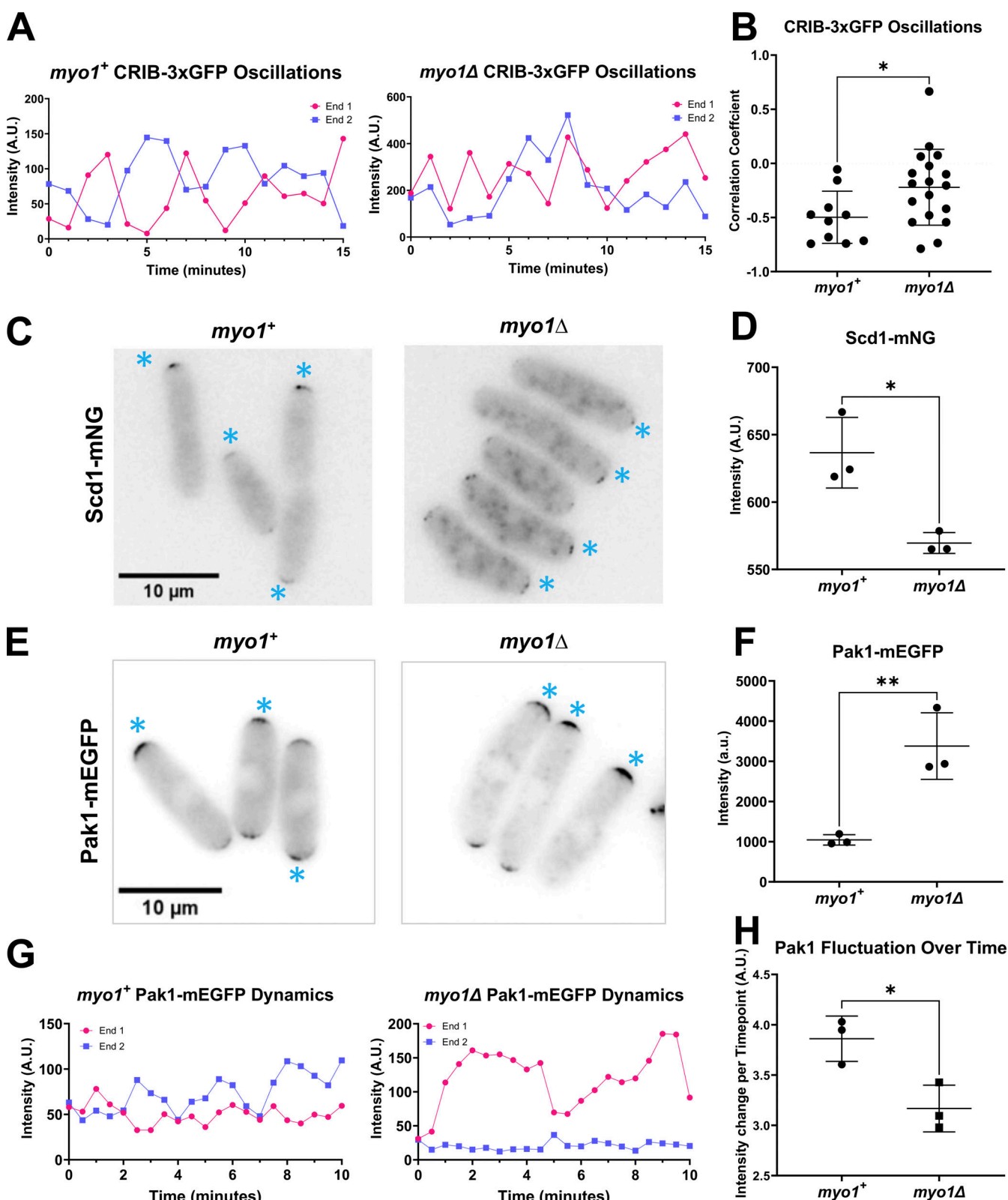

Figure 7. **Endocytic mutant *myo1Δ* cells show disrupted Cdc42 activation dynamics similar to CK-666 treatment. (A)** CRIB-3xGFP dynamics were observed in *myo1+* and *myo1Δ* cells. **(B)** Correlation coefficients of active Cdc42 oscillations between cell ends in *myo1+* cells and *myo1Δ* mutants (*n* ≥ 10 cells). **(C)** Scd1-mNG localization in *myo1+* and *myo1Δ* cells. Asterisks indicate Scd1-mNG localization at the brighter cell end. **(D)** Quantification of Scd1-mNG accumulation at cell ends in *myo1+* and *myo1Δ* cells (*N* = 3, *n* ≥ 10 cells). **(E)** Pak1-mEGFP localization in *myo1+* and *myo1Δ* cells. Asterisks indicate Pak1-mEGFP localization at the brighter cell end. **(F)** Quantification of Pak1-mEGFP accumulation at cell ends in *myo1+* cells and *myo1Δ* mutants (*N* = 3, *n* ≥ 10 cells). **(G)** Pak1-mEGFP dynamics at cell ends in *myo1+* and *myo1Δ* cells. **(H)** Quantification of the extent of Pak1-mEGFP fluctuation in *myo1+* and *myo1Δ* cells (*N* = 3, *n* ≥ 10 cells). Scale bar, 10 µm. P value, *<0.05, **<0.0085, Student's *t* test.

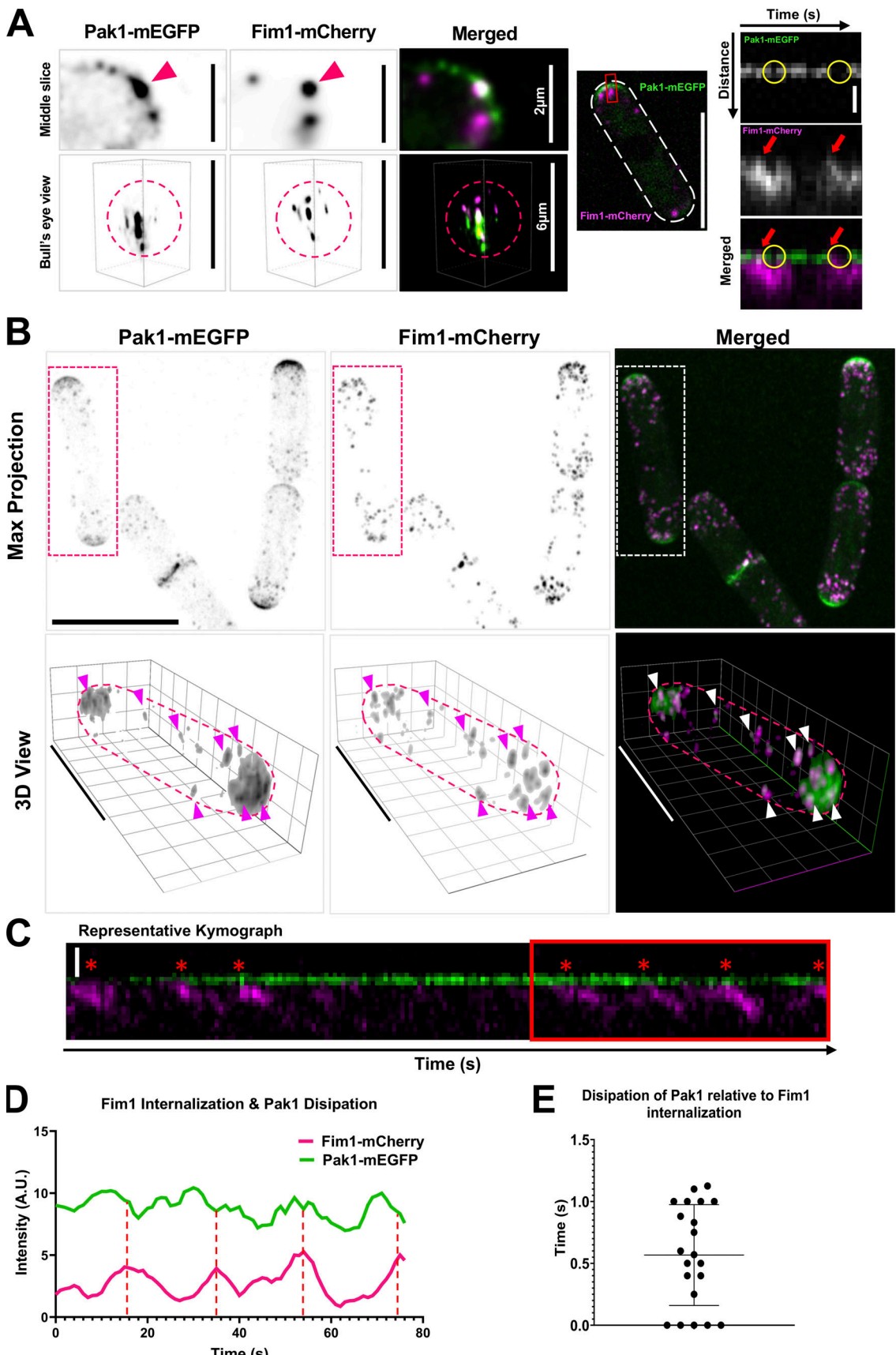

Figure 8. **Pak1 removal from the plasma membrane is associated with endocytic patch internalization. (A)** Fim1-mCherry and punctate Pak1-mEGFP localization at cell ends. Middle slice shows a single frame of a cell end expressing Pak1-mEGFP and Fim1-mCherry. Arrowhead marks overlapping Fim1-

mCherry and Pak1-mEGFP puncta (scale bar = 2 µm). Bull's eye view shows 3D reconstructed image of the same cell end. A dashed circle marks the outline of the cell end (scale bar = 6 µm). Center panel shows a whole cell with a white dashed outline (scale bar = 10 µm). Red box indicates the region shown as kymographs in the right panels. Yellow circles mark Pak1-mEGFP loss from the membrane while red arrows mark the onset of Fim1-mCherry internalization at the membrane (scale bar = 800 nm). Kymograph brightness is adjusted for ease of visibility for changes in Pak1 intensity. Representative images of the cell end in the left panel are deconvolved, clarified, and denoised with Nikon NIS elements. **(B)** Super-resolution images of cells expressing Pak1-mEGFP and Fim1-mCherry using Airyscan microscopy. Top row shows a maximum intensity projection. The dotted ROI indicates the cell that is presented in 3-D below). The bottom row shows 3D view of Pak1-mEGFP and Fim1-mCherry localization in the indicated cell. Magenta outline indicates the location of the cell (scale bar = 10 µm). Arrows indicate examples of Pak1-mEGFP at Fim1-mCherry patches. Airyscan images are processed and presented in 3D using Zeiss Zen Blue Automatic processing and 3D deconvolution. **(C)** Kymograph of Pak1-mEGFP and Fim1-mCherry as captured in a 3-min timelapse movie with 1-s intervals. Red box marks the region quantified in C (scale bar = 800 nm; frame rates = 1 s per frame [SPF]). **(D)** Quantification of Pak1-mEGFP and Fim1-mCherry intensities at the membrane over time. Red dashed lines indicate the peak and subsequent drop of Fim1-mCherry intensity. **(E)** Quantification of the time taken for dissipation of Pak1-mEGFP relative to Fim1-mCherry internalization ($n \geq 6$ endocytic events per kymograph, $N = 21$ kymographs from 3 replicates).

and Verde, 2013; Pelham and Chang, 2002; Wu and Lew, 2013). However, the molecular details of how these proteins undergo oscillatory dynamics are not clear. At each cell end, Cdc42 activation occurs via positive feedback and time-delayed negative feedback, resulting in the oscillatory pattern (Das et al., 2012, 2015; Wu and Lew, 2013; Xu and Jilkine, 2018). The molecular details of how each cell end overcomes this negative feedback to allow reactivation of Cdc42 at that end is not known. Here, we found that endocytosis enables periodic reactivation of Cdc42 at the cell ends by removing the negative feedback and allowing the Cdc42 GEFs to return to that end.

Arp2/3-dependent endocytosis is known to be required for growth in *S. pombe* and other fungal species as well as mammalian cells with high membrane tension (Aghamohammadzadeh and Ayscough, 2009; Basu et al., 2014; Boulant et al., 2011; Epp et al., 2010). Previous works show that Arp2/3-dependent endocytosis is required to overcome the high internal turgor pressure for proper polarization and growth in yeasts (Aghamohammadzadeh and Ayscough, 2009; Basu et al., 2014). Here, we expound upon the importance of Arp2/3-dependent endocytosis for proper spatiotemporal regulation of polarity factors. We show that adequate removal of inhibitors is critical for proper polarized growth. Indeed, our lab has previously reported the importance of Rga4 removal for proper cell cycle progression and resumption of growth (Rich-Robinson et al., 2021). While previous works have described how exocytosis dilutes polarity protein concentration and dampens the concentration of active Cdc42 at that site, our work finds that endocytosis promotes Cdc42 activity through the removal of inhibition (Ghose and Lew, 2020; Watson et al., 2014). Interestingly, we also find that Pak1 is required for normal endocytic patch dynamics. This agrees with our previous results that show a role for Cdc42 in regulating endocytosis (Campbell et al., 2022; Onwubiko et al., 2019). Pak1 is a potential kinase for the activation of the type 1 myosin Myo1 (Attanapola et al., 2009; Wu et al., 1997). The role of Cdc42 and Pak1 kinase in endocytosis is not well understood. While genetic data suggest Cdc42 and Pak1 in the regulation of endocytosis, the mechanism of this process has not been elucidated. Furthermore, our data does not describe how Pak1 is removed from the membrane via endocytosis. The Cdc42 GEF Gef1 localizes to the endocytic patches in an F-BAR protein Cdc15-dependent manner (Hercyk and Das, 2019). Cdc15 has been shown to bind Myo1 at these patches (Arasada and Pollard, 2011; Arasada et al., 2018; Carnahan and Gould, 2003).

Thus, Pak1 may be part of a protein complex including its substrate Myo1, Cdc42, Gef1, and Cdc15. Both Cdc15 and Myo1 disappear from the membrane upon patch internalization (Arasada and Pollard, 2011). It is possible that Pak1 is lost along with that complex. Further investigations will demonstrate the molecular details of these regulations.

The time-delayed negative feedback in Cdc42 regulation is mediated by its effector kinase Pak1 (Das et al., 2012; Howell et al., 2012; Ottilie et al., 1995; Rapali et al., 2017; Tu and Wigler, 1999; Wu and Lew, 2013). Here, we show that impeding endocytosis either by inhibition of the Arp2/3 complex or by deletion of *myo1* results in Pak1 accumulation at the cell ends. This suggests that endocytosis promotes Pak1 removal from the cell ends. Without Pak1 removal, Scd1 cannot localize to activate Cdc42, and, without further Cdc42 activation, the cell ends fail to compete for resources. Pak1 removal enables Scd1 localization to the cell ends and allows for proper Cdc42 activation dynamics between the two cell ends. Thus, endocytosis is required for maintaining anticorrelated Cdc42 activation dynamics between the cell ends, thus promoting bipolar growth. Previous models have demonstrated the need for negative regulation of active Cdc42 for its oscillations (Das et al., 2012; Das and Verde, 2013; Howell et al., 2012; Xu and Jilkine, 2018). However, these models did not explore the impact of endocytosis on active Cdc42 dynamics and cell polarity. In this study, we developed models to illustrate how the accumulation of active Cdc42 hinders the buildup of Scd1 through the excess accumulation of Pak1 on the cellular membrane. Additionally, we found that endocytosis allows cellular bipolarity by facilitating the removal of Pak1.

While active Cdc42 is required for polarized growth, it is possible to have too much of a good thing. The absence of negative feedback, due to the loss of Pak1 activity, leads to an abundance of positive feedback at the dominant end. Thereby, Cdc42 activation at the dominant end is too strong for the second end to compete against, resulting in aberrant rounded morphology and monopolar growth as observed in *pak1-ts* mutants (Das et al., 2012; Sawin et al., 1999). We found that inhibition of the Arp2/3 complex due to CK-666 treatment or *myo1Δ* leads to enhanced negative feedback due to the stabilization of Pak1 at the cell ends. This enhanced negative feedback results in monopolar Cdc42 activity and growth is inhibited at the site where Pak1 stabilizes. Positive feedback leads to Cdc42 activation which, in turn, allows Pak1 activation and time-delayed negative feedback. We reported that Pak1 plays a role in promoting

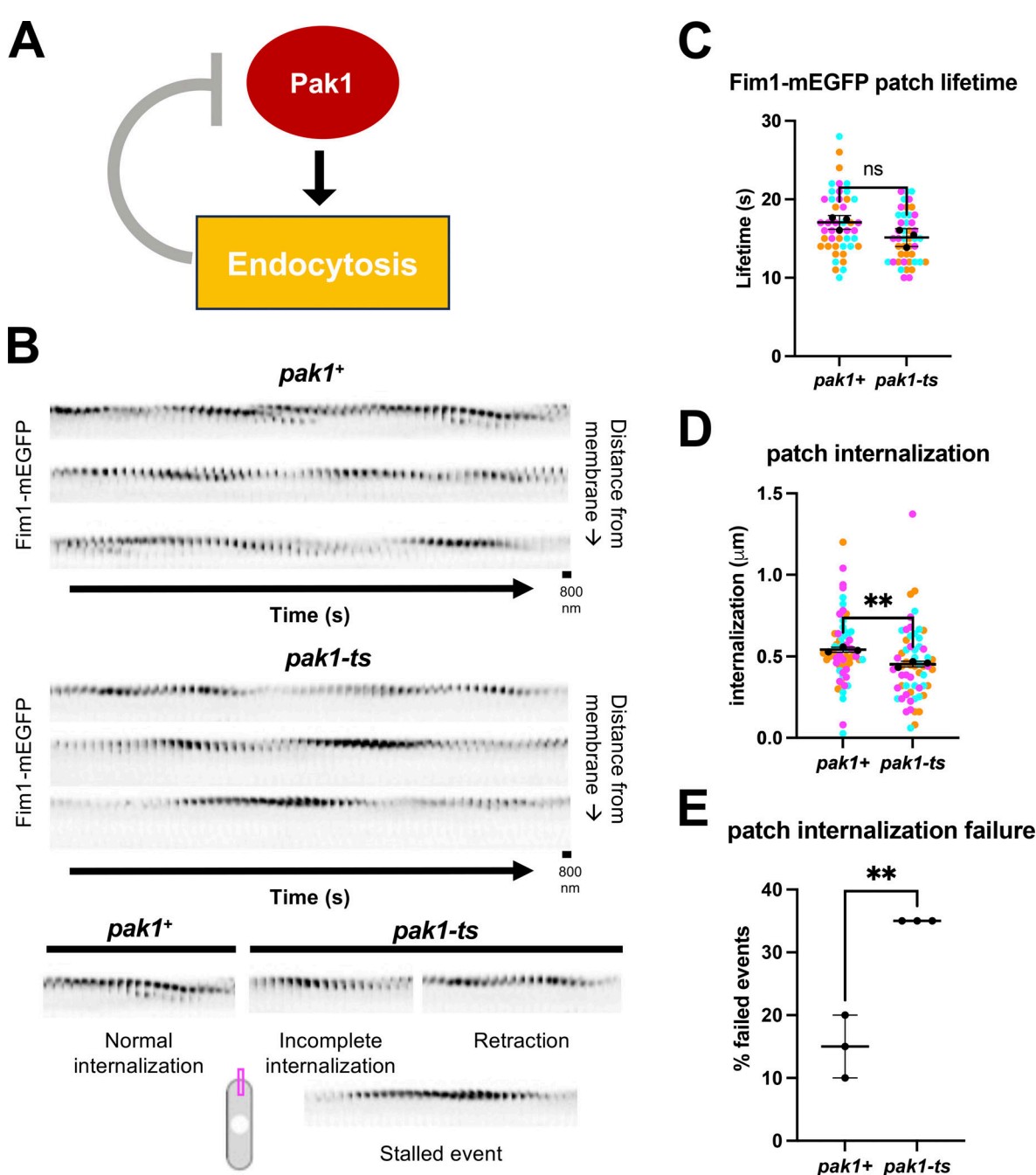

Figure 9. **Pak1 kinase plays a role in promoting endocytosis. (A)** Schematic depicting our hypothesis that Pak1 activity promotes its own removal through endocytosis. **(B)** Montages of Fim1-mEGFP labeled endocytic patches at growing ends of interphase *pak1+* and *pak1-ts* cells (scale bars = 800 nm; frame rates = 1 s per frame [SPF]). **(C)** Quantification for lifetimes of Fim1-mEGFP labeled endocytic patches in *pak1+* and *pak1-ts* (n = 15 endocytic patches per experiment, N = 3 experiments). **(D)** Quantification of Fim1-mEGFP labeled endocytic patch internalization into the cell interior from the plasma membrane in *pak1-ts* mutants compared to *pak1+* cells (n = 20 endocytic patches per genotype per experiment, N = 3 experiments). **(E)** Quantification of failed endocytic events in *pak1+* cells compared to *pak1-ts*. Failed events do not internalize beyond 350 nm from the plasma membrane (n = 20 endocytic patches per genotype per experiment, N = 3 experiments). Symbol colors in graphs = distinct experiments. Solid symbols = means of experiments. n.s., not significant; P value, **<0.01, Student's *t* test.

endocytosis (Fig. 9). Our findings show that disruption of negative feedback via endocytosis-dependent Pak1 removal enables the return of positive feedback to the same cell end.

*pak1-ts* cells are characteristically monopolar. However, we observe that *pak1-ts* cells can localize Scd1 in a bipolar manner in a fraction of the cells, upon CK-666 treatment. We hypothesize

that this may be explained by a secondary form of negative regulation mediated by the Arp2/3-complex that is still unknown. As per Model 3, bipolar Scd1 in the absence of Pak1 occurs when Scd1 at the end is stabilized. It is possible that endocytosis partially contributes to Scd1 removal by an unknown mechanism and this is observable only in the absence of Pak1

kinase. Further research will investigate the mechanism of this second negative regulation. One possibility could be that endocytosis leads to changes in the levels of proteins at the cortex thus altering Cdc42 activation. Alternately, endocytosis may alter the lipid composition of the plasma membrane thereby affecting Cdc42 activation.

While the functions and localizations of Cdc42 and its regulators have been extensively studied, the spatiotemporal localization (phase shifts) among these proteins has not been shown in vivo. Previous work has elucidated phase shifts among polarity factors, ion signaling, and growth in pollen tubes (Hwang et al., 2005; Messerli and Robinson, 1997; Messerli et al., 1999). In vitro experiments using *Xenopus* frog egg extracts and protein-reconstituted systems have also shown phase shifts between waves of Rho GTPase activity and actin polymerization (Bement et al., 2015; Landino et al., 2021). Our data show that the accumulation of active Cdc42 is in phase with the recruitment of its activator Scd1 but, by contrast, the accumulation of Pak1 shows a phase delay relative to the positive regulators. The phase shift between Pak1 and the positive regulators likely results from its multiple roles and step-wise regulation of both Pak1 kinase and active Cdc42. In vitro investigations show that active Cdc42 physically binds and activates Pak1 kinase at the plasma membrane (Ottilie et al., 1995; Rapali et al., 2017; Tu and Wigler, 1999). However, the dynamics of Cdc42-dependent Pak1 recruitment at the cell ends has not been fully investigated. These phase shifts suggest that Pak1 kinase can remain at the membrane at least for a small period of time after the loss of active Cdc42. Indeed, we found that Pak1 dynamics at the membrane are slower than that of the Cdc42 activators Scd1 and Scd2. It is unknown if Pak1 once recruited still needs to or even can keep binding active Cdc42 once it interacts with downstream substrates. Further investigations will explore the nature of these interactions and dynamics. Overall, our data suggests that the inhibitory role of increasing Pak1 activity causes a gradual decrease in its own recruitment until, eventually, the Pak1 removal rate exceeds the recruitment rate. This intrinsic delay is further evidence of the complexity of self-organized processes and the multiple mechanisms in cell polarity.

Self-organization is seen throughout biology as a means to precisely regulate cellular processes (Karsenti, 2008). Our work furthers the understanding of Cdc42's extensive self-organizing capabilities (Gerganova et al., 2021; Lamas et al., 2020a; Rutkowski et al., 2023, *Preprint*). Our findings show that the Cdc42 regulatory complex along with actin-mediated endocytosis forms a self-organizing unit in the regulation of cell polarity. Complex signaling pathways with higher-order molecular feedbacks have been observed in several processes and systems (Glazenburg and Laan, 2023; Guo and Dong, 2022; Karsenti, 2008; Landge et al., 2020; Stock and Pauli, 2021). In nature, robustness is important but cannot come at the expense of adaptability or vice versa. These higher-order pathways allow for adaptability while maintaining robustness. The RHO family GTPases undergo higher-order pathways via multiple feedback loops and this allows for their precise spatiotemporal activation necessary for cell polarization (Fairn et al., 2011; Hedrick et al., 2016; Lamas et al., 2020a; Martin et al., 2016; Martin-Vilchez

et al., 2017; Nobes and Hall, 1995). Our findings on Cdc42 regulation define higher-order signaling pathways with multiple feedback loops which ensure polarized growth at each cell end and allow for bipolarity. While the role of Cdc42 regulating membrane trafficking is well documented, here, we show how membrane trafficking also contributes to Cdc42 regulation and maintenance of bipolar growth.

## Materials and methods
### Strains and cell culture
The *S. pombe* strains used in this study are listed in Table S3. All strains are isogenic to the original strain PN567. Cells were cultured in yeast extract (YE) medium and grown exponentially at 25°C unless specified otherwise. Standard techniques were used for genetic manipulation and analysis (Moreno et al., 1991). Cells were grown exponentially for at least three rounds of eight generations each before imaging.

### Microscopy
Imaging was performed at room temperature (23–25°C). We used an Olympus IX83 microscope equipped with a VTHawk two-dimensional array laser scanning confocal microscopy system (Visitech International), a Hamamatsu electron-multiplying charge-coupled device digital camera (Hamamatsu EM-CCD Digital Camera ImageM Model: C9100-13 Serial No: 741262), and a 100×/1.49 NA UAPO lens (Olympus). Images were acquired using MetaMorph (Molecular Devices). This microscope was used for the acquisition of CRIB-3xGFP oscillations (Fig. 1).

We also used a spinning disk confocal microscope system with a Nikon Eclipse inverted microscope with a 100×/1.49 NA lens, a CSU-22 spinning disk system (Yokogawa Electric Corporation), and a Photometrics EM-CCD camera (Photometrics Technology Evolve with excelon Serial No: A13B107000). Images were acquired using MetaMorph (Molecular Devices). This system was used to test the impact of CK-666 on Cdc42 regulators (Fig. 2), the localization of Pak1-mEGFP (Fig. 4, A, C, and E), and the impact of CK-666 on Scd1-mNG localization in *pak1-ts* mutants (Fig. 5, A and B).

Additionally, we used a Nikon Ti2 Eclipse wide-field microscope with a 100×/1.49 NA objective and an ORCA-FusionBT digital camera (Hamamatsu Model: C15440-20UP Serial No: 500428). Images were acquired using Nikon NIS Elements (Nikon). Fluorophores were excited using an AURA Light Engine system (Lumencor). This system was used to capture Pak1-mEGFP dynamics (Fig. 4, D and F; and Fig. S5 A), the phase shifts between polarity proteins (Fig. 6, B–D; and Fig. S3, A and B), the localization and dynamics of CRIB-3xGFP, Scd1-mNG, and Pak1-mEGFP in *myo1Δ* mutants (Fig. 7 and Fig. S5 C), and the punctate localization of Pak1-mEGFP at cell ends (Fig. 8 B, leftmost panel).

Microscopy was also performed with a 3i spinning disk confocal using a Zeiss AxioObserver microscope with an integrated Yokogawa spinning disk (Yokogawa CSU-X1 A1 spinning disk scanner) and a 100×/1.49 NA objective. Images were acquired with a Teledyne Photometrics Prime 95b back-illuminated sCMOS camera (Serial No: A20D203014). Images

were acquired using SlideBook (3i Intelligent Imaging innovations). This system was used to test if loss of branched actin alone causes a stress response (Fig. S1), for FRAP analysis (Fig. 4 B, 6 F, S3 C, and S5 B), the dissipation of Pak1-mEGFP relative to Fim1 internalization (Fig. 8 A, middle and right panels, Fig. 8, C–E), and the role of Pak1 in promoting endocytosis (Fig. 9).

Additional imaging was performed using a Zeiss LSM 880 using Airyscan and a Plan-APOCHROMAT 63×, 1.4 NA oil objective at a 5.0 optical zoom and 0.4-µm step size through the entire depth of cells using Zeiss Zen Black software. Postprocessing of Airyscan images was completed in Zeiss ZEN Blue 2.3 using automatic processing and 3-D deconvolution. This system was used to investigate if Pak1-mEGFP localizes with Fim1-mCherry patches (Fig. 8 B).

### Acquiring and quantifying fluorescence intensity
Fluorescence intensity was measured using ImageJ software. All images were sum projected and mean intensities were reported. Freehand ROIs were used to measure the signal at the cell ends. For analysis of two ends of the same cell, cytoplasm with little or no signal was used for background subtraction. For all other experiments, a region outside of the cell was used for background subtraction. The mean intensity was measured and recorded after the background subtractions.

To quantify the anticorrelation between the two ends, cells in each condition were imaged every minute for 60 min. The data from each cell was then analyzed to find the correlation coefficient between fluorescent signals of both ends within the cell.

To quantify signal stability at the cell ends, cells were imaged every 30 s for 20 min. The stability of the proteins was then quantified by measuring the difference in intensity between each subsequent time point for the duration of the timelapse imaging. Populations that had larger average variations of fluorescent intensity between each timepoint were more dynamic and less stable than populations that had smaller variations between each timepoint.

To observe and quantify the in vivo phase shifts, cells were imaged every 6 s for 9 min. Two fluorescent proteins were imaged simultaneously using the 488 and 561 nm wavelengths. LED power was set to 3% for each wavelength. Fluorescence intensity for each protein was measured for each timepoint to indicate how the signal is changing over time. The phase lag between each protein at each cell end was then quantified using crosscorrelation analysis wherein the correlation coefficient is calculated for both signals with and without shifting the quantified signal. The second fluorescent signal was shifted backward by 20 time points while the first signal remained the same, and the correlation coefficient was calculated for each shift. The correlation coefficient was also calculated as the signal was shifted forward by 20 time points and calculated for each shift. The number of shifts required to achieve the best correlation coefficient indicates how far one signal lags the other. The shift required to produce the best correlation between the two fluorescent signals was verified as showing true correlation or not before being included in the final comparative analysis for each set of protein signals.

For FRAP analysis, cells were imaged every 250 ms for 30 s with 15% laser power at 488 nm wavelength. Square 10 × 10 pixel (1.1 × 1.1 µm) ROIs were used to target, bleach, and measure the fluorescence intensity at half of one cell end as it was bleached and then allowed to recover. The first nine time points were acquired before photobleaching and served as a measurement of the initial fluorescence intensity. On the 10th time point, the acquisition was halted momentarily and the region of interest was photobleached using three 4% laser power bleach repetitions with a duration of 5 ms each (15 ms total). After the region of interest was bleached, acquisition resumed at 250 ms intervals as the fluorescence intensity recovered. The half-life of fluorescence recovery was quantified using SlideBook FRAP analysis software.

To observe Pak1-mEGFP and Fim-mCherry from the bull's eye view, triggered acquisition was used to simultaneously image cells using the Nikon widefield microscope at 488 and 561 nm wavelengths at 5% LED power. Cells were imaged with 31 Z-sections at 0.2-µm step size through the depth of the cell for a total distance of 6 µm. Images were then clarified, denoised, and 3-D deconvolved in Nikon NIS elements.

To verify if Pak1-mEGFP localization with Fim1-mCherry patches, super-resolution Airyscan microscopy was used. Cells were imaged at 488 nm with 10% laser power and 561 nm at 2% laser power. 16 Z-sections were captured with 0.4-µm step size over a total depth of 4.5 µm. Images were then automatically processed and 3-D deconvolved in Zeiss Zen Blue software.

To quantify the loss of Pak1 from the cell membrane in relation to the internalization of Fim1, cells were imaged each second for 3 min. Rectangular ROIs were used to isolate regions of the cell end for analysis. These ROIs were used to make kymographs of both the Fim1-mCherry signal and the Pak1m-EGFP signal. Kymographs were quantified by plotting the intensity profile of both proteins at the cell membrane. The intensity profiles were then analyzed by recording when Pak1-mEGFP intensity began to decrease in comparison to each successful endocytic event. Between 6 and 10 endocytic events were captured in each kymograph. These events were then averaged for each kymograph and graphed.

### Actin cytoskeleton disruptions
Cells were treated with 100 µM CK-666 (SML006-5MG; Sigma-Aldrich) in DMSO (D8418-250ML; Sigma-Aldrich) to block the Arp2/3 complex and branched actin assembly. To block the polymerization of all F-actin structures, cells were treated with 10 µM Latrunculin A (LatA; EMD Millipore) dissolved in DMSO for 30 min prior to imaging. Control cells were treated with 0.1% DMSO in YE media.

### Statistical tests
Significance was determined using GraphPad Prism. When comparing two conditions, a Student's *t* test was used (twotailed, unequal variance). One-way ANOVA, followed by Tukey's multiple comparison test, was used to determine significance for experiments with three or more conditions. SuperPlots (Lord et al., 2020) were made using GraphPad Prism.

## Mathematical models for regulation networks

The details of the regulatory network are provided in the supplementary materials. This paper introduces three networks, all described using a reaction–diffusion model based on the Xu-Jilkine model (Xu and Jilkine, 2018). Model 1 and Model 2 each include one negative regulatory pathway of active Cdc42. In Model 1, there is a direct inhibition from active Cdc42 to Scd1. In Model 2, the negative pathway involves Pak1. Model 3, on the other hand, includes two negative regulatory pathways of active Cdc42, with one involving Pak1. However, the details of the second pathway remain unclear. For the numerical analysis, MATLAB was used.

## Computational phase analysis

Z-scores were used to normalize the data. Prior to analysis, the normalized data was smoothened, and the Hilbert Transform of the data was applied using MATLAB.

## Online supplemental material

The modeling supplement shows equations, parameter values, and detailed assumptions of the mathematical models. Fig. S1 shows that loss of all F-actin structures causes a stress response but CK-666 treatment alone does not. Fig. S2 shows the impact of reduced detachment rates on untreated bipolar and CK666-treated monopolar dynamics. Fig. S3 shows montages of phase shift between pairs of polarity proteins. Fig. S4 shows the reduction and elimination of phase shift between Cdc42-GTP and Pak1 with increased Pak1 attachment/detachment rate constants in the model. Fig. S5 shows Pak1-mEGFP and CRIB-GFP dynamics under different conditions with CK666. Pak1-mEGFP appears to stabilize at the ends upon CK666 treatment. Video 1 shows active Cdc42 oscillations in a DMSO-treated *S. pombe* cell. Active Cdc42 probe CRIB-3xGFP oscillates at the cell ends in an interphase cell treated with DMSO. Imaged with a laser scanning confocal microscopy system in 1-min intervals for 1 h. Displayed at seven frames per second. Supplement for Fig. 1 A. Video 2 shows active Cdc42 oscillations in a CK-666 treated *S. pombe* cell. Oscillations at the cell ends of the active Cdc42 probe CRIB-3xGFP are disrupted in cells treated with 100 μm CK666. Imaged with a laser scanning confocal microscopy system in 1-min intervals for 1 h. Displayed at 7 frames per second. Supplement for Fig. 1 A. Video 3 shows Pak1 kinase oscillations in a DMSO-treated *S. pombe* cell. Pak1-mEGFP displays oscillatory behavior at the cell ends in interphase cells. Imaged with time-lapse epifluorescence microscopy in 30 s intervals for 20 min. Displayed at seven frames per second. Supplement for Fig. S5 A. Video 4 shows Pak1 kinase oscillations in a CK-666-treated *S. pombe* cell. Pak1-mEGFP oscillatory behavior is disrupted in cells treated with 100 μm CK666. Pak1-mEGFP appears to stabilize at the ends upon CK666 treatment. Imaged with time-lapse epifluorescence microscopy in 30-s intervals for 20 min. Displayed at seven frames per second. Supplement for Fig. S5 A. Table S1 shows parameter values of the wild-type model2, including parameters in model2-1. Table S2 shows parameter set comparisons. Table S3 shows the strain list. Data S1 shows data for the plots shown in the figures of the article. Each figure panel corresponds to a different sheet.

## Data availability

The data underlying all in vivo figures are available in the published article and its online supplemental material (Data S1). The computer code for the model is available in the online supplemental material.

## Acknowledgments

We thank James Moseley and Vladimir Sirotkin for providing strains, and Bret Judson at the Boston College Imaging Core for imaging support. We thank Guangyuan Liao for helping with the MATLAB code.

This work is supported by the National Institutes of Health (NIH) grant R01GM136847 to M. Das. T. Hong is supported by NIH grant R35GM149531. Open Access funding provided by Boston College.

Author contributions: M.A. Harrell: Conceptualization, Formal analysis, Investigation, Validation, Visualization, Writing - original draft, Writing - review & editing, Z. Liu: Conceptualization, Formal analysis, Investigation, Methodology, Software, Validation, Visualization, Writing - review & editing, B.F. Campbell: Formal analysis, Investigation, Validation, Visualization, Writing - review & editing, O. Chinsen: Investigation, T. Hong: Conceptualization, Funding acquisition, Resources, Supervision, Writing - original draft, Writing - review & editing, M. Das: Conceptualization, Funding acquisition, Methodology, Project administration, Resources, Supervision, Visualization, Writing - review & editing.

Disclosures: The authors declare no competing interests exist.

Submitted: 21 November 2023

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

# Supplemental material

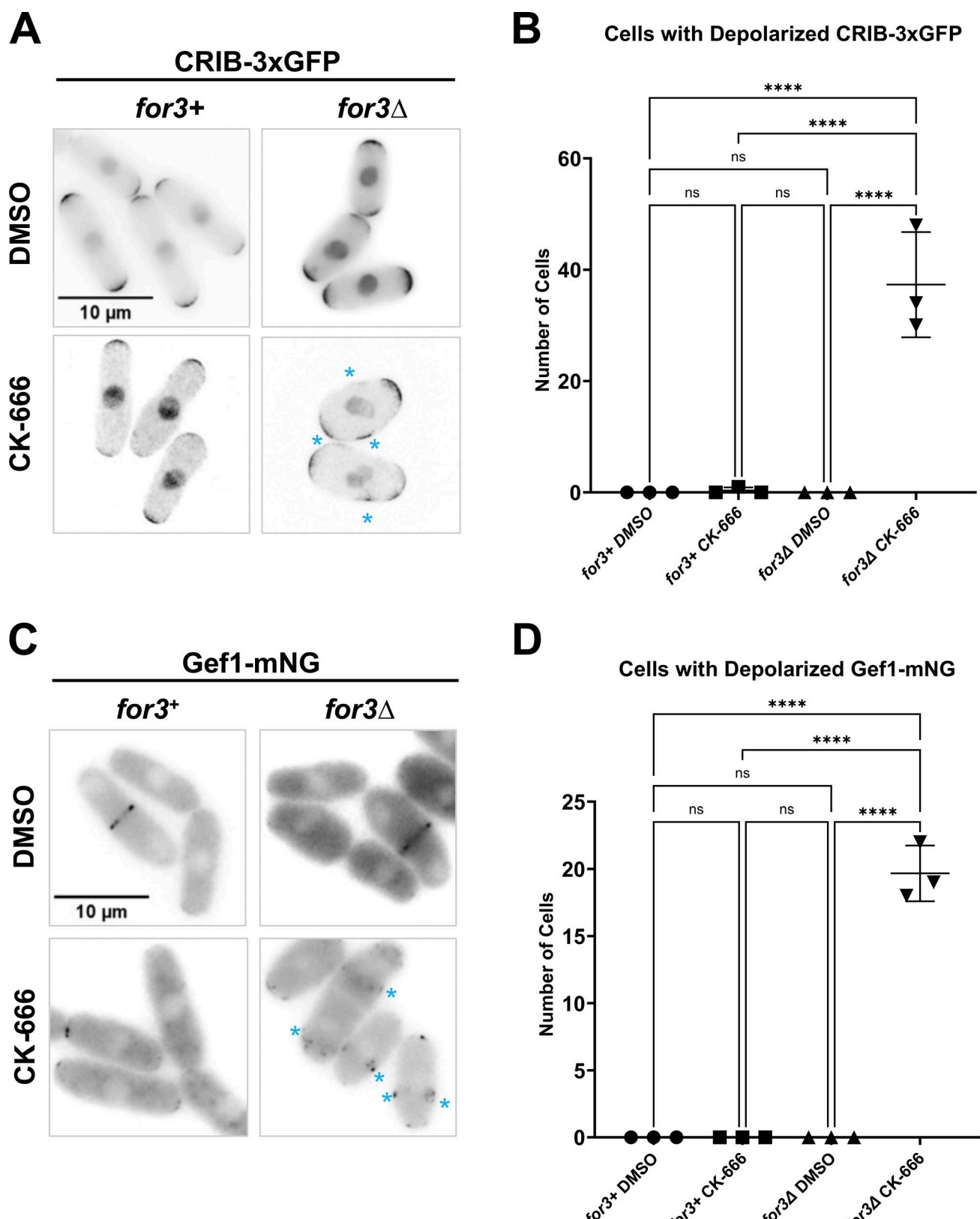

Figure S1. **Loss of all F-actin structures causes a stress response but CK-666 treatment alone does not. (A)** CRIB-3xGFP localization in *for3+* and *for3Δ* cells treated with DMSO and CK-666. **(B)** Quantification of the number of cells with depolarized CRIB-3xGFP localization. **(C)** Gef1-mNG localization in *for3+* and *for3Δ cells* treated with DMSO and CK-666. **(D)** Quantification of the number of cells with depolarized Gef1-mNG localization. Scale bar, 10 µm. n.s., not significant; P value, ****<0.0001, one-way ANOVA, followed by Tukey's multiple comparison test.

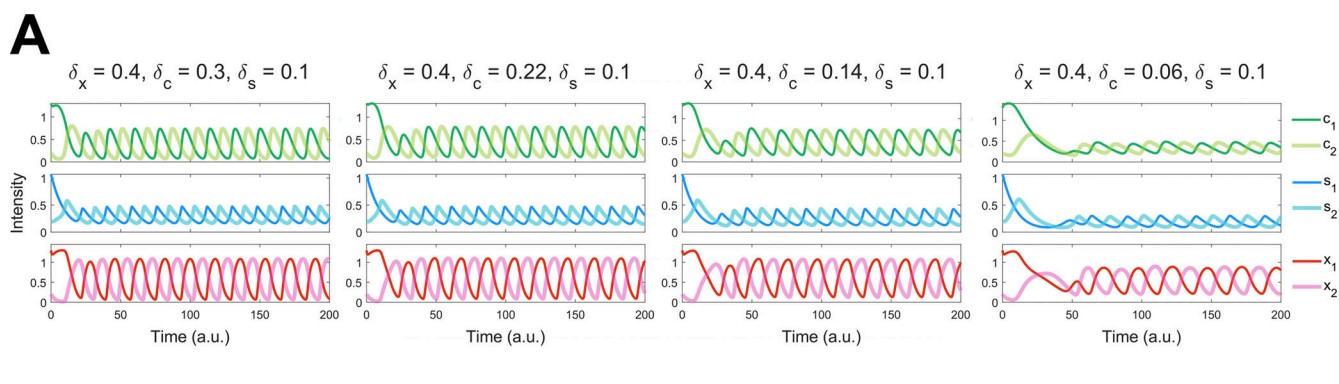

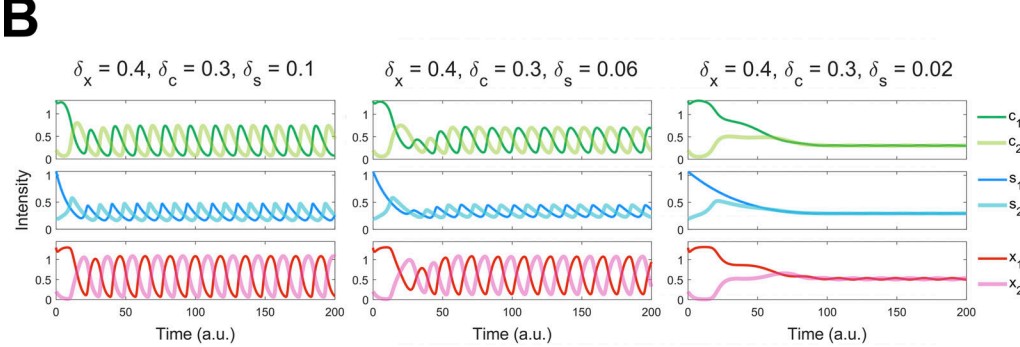

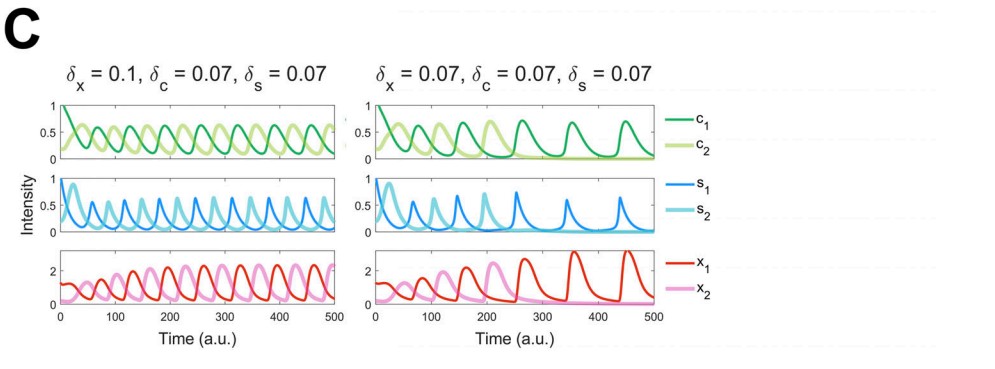

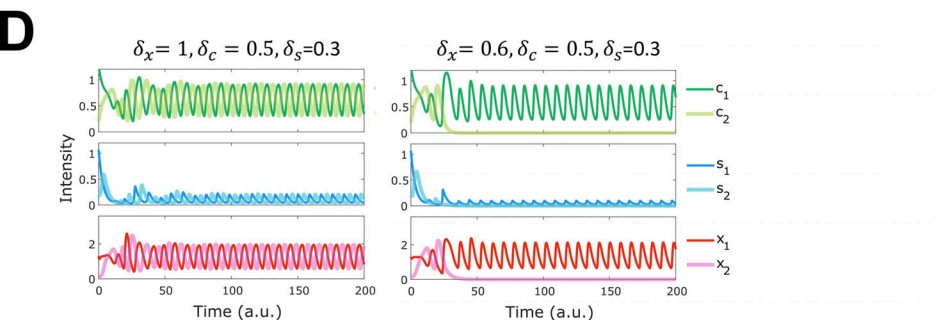

Figure S2. **Impact of reduced detachment rates on untreated bipolar and CK666-treated monopolar dynamics.** Bipolar dynamics were preserved with varied detachment rates of active Cdc42 and Scd1. **(A)** Bipolar dynamics after reducing the detachment rate of Cdc42 to 20% of the wild-type detachment rate. **(B)** Protein dynamics upon reducing the detachment rate of Scd1 to 20% of the wild-type rate. **(C)** The second parameter set effectively fits the untreated bipolar and CK666-treated monopolar data in Model 2. $\delta_x = 0.1$, $\delta_c = 0.07$, $\delta_s = 0.07$ for untreated, $\delta_x = 0.07$, $\delta_c = 0.07$, $\delta_s = 0.07$ for CK666-treated. **(D)** The third parameter set fits the untreated bipolar and CK666-treated monopolar in Model 2. $\delta_x = 1$, $\delta_c = 0.5$, $\delta_s = 0.3$ for untreated, $\delta_x = 0.6$, $\delta_c = 0.5$, $\delta_s = 0.3$ for CK666 condition.

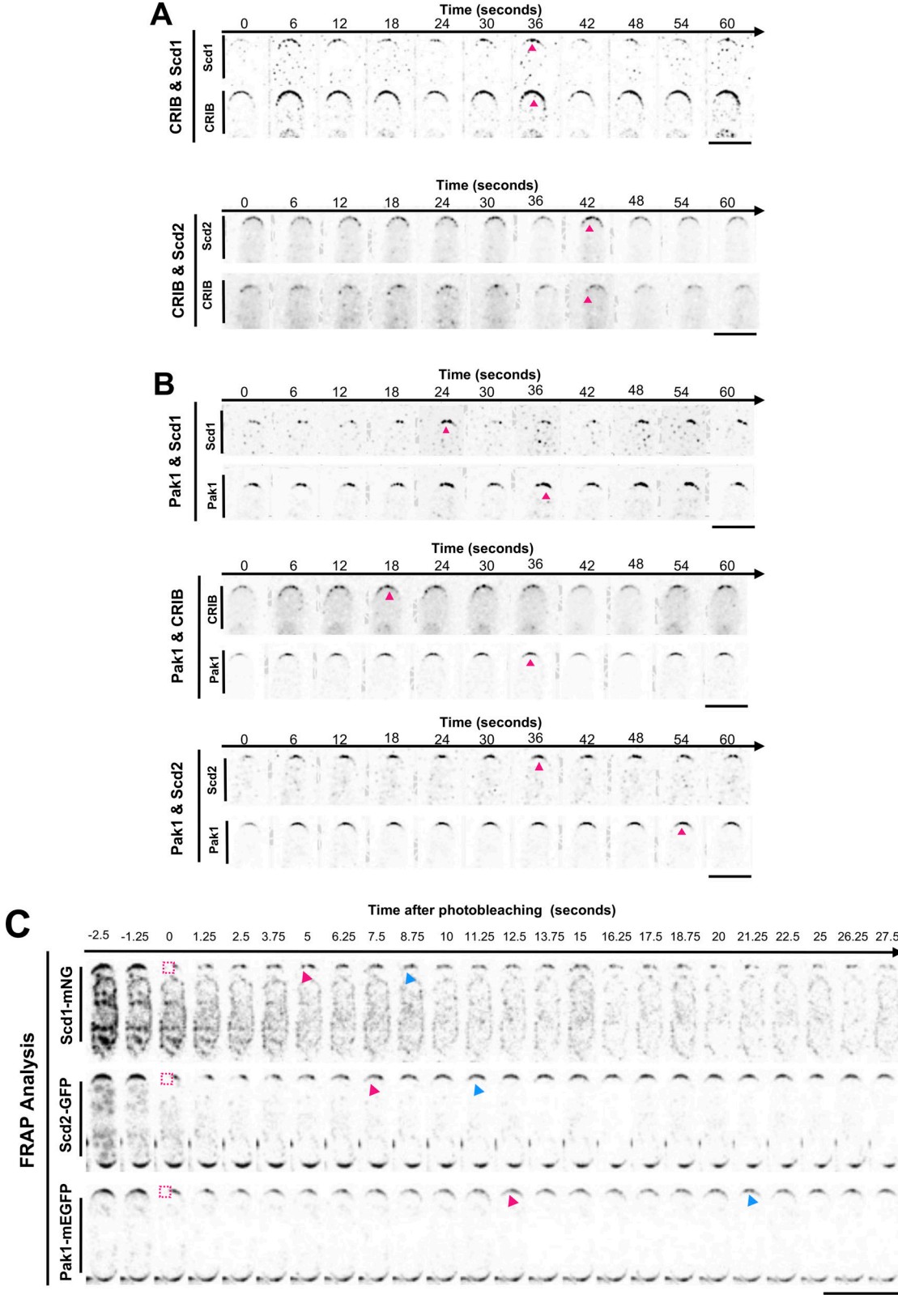

Figure S3. **Montages of phase shift between pairs of polarity proteins. (A)** Montage of phase shift between Scd1-tdTomato and Cdc42 activity (CRIB-3xGFP) and between Scd2-mCherry and CRIB-3xGFP. Arrows denote peaks in intensity. **(B)** Montage of phase shift between Pak1-mEGFP and the positive feedback proteins: Scd1-tdTomato, active Cdc42 (CRIB-mCherry), and Scd2-mCherry. Arrows indicate peaks in intensity Scale bar, 5 µm. **(C)** FRAP analysis of Scd1-mNG, Scd2-GFP, and Pak1-mEGFP recovery after photobleaching half the cell end. Magenta boxes indicate the half of the cell end that has been photobleached. Magenta arrows show when half of the final fluorescence has recovered. Blue arrows indicate when the fluorescence recovery plateaus. Scale bar 10 µm. Phase shift montages clarified and denoised, made in Nikon NIS elements.

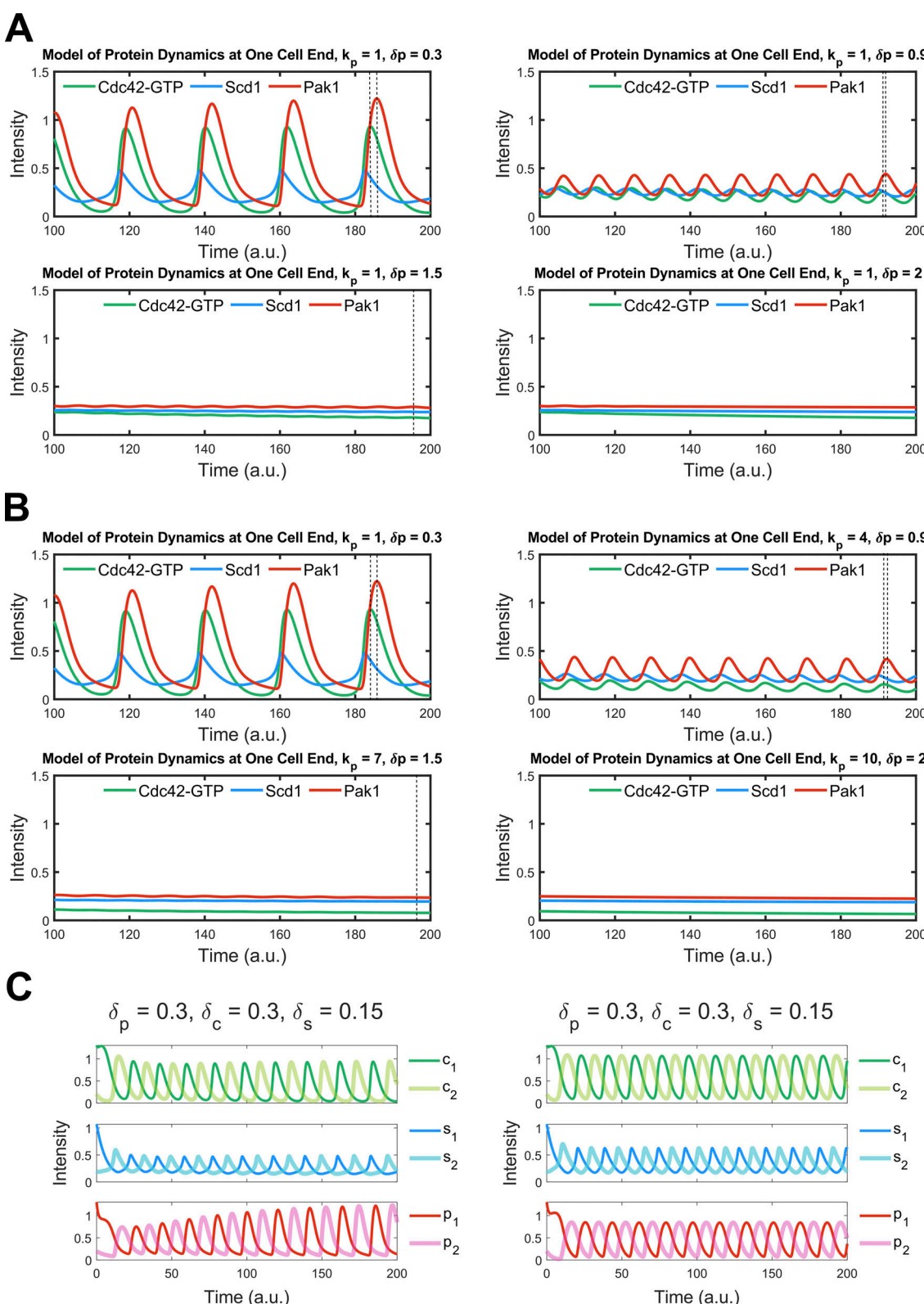

Figure S4. **Reduction and elimination of phase shift between Cdc42-GTP and Pak1 with increased Pak1 attachment/detachment rate constants in the model. (A)** Incrementally increasing the detachment rate constant of Pak1 ($\delta_p$) from 0.3 to 2 leads to a gradual reduction in the phase shift between Cdc42-GTP and Pak1. As $\delta_p$ nears 2, this shift becomes less pronounced, culminating in dampened oscillations that eventually subside. **(B)** A simultaneous elevation of both the attachment rate constant ($k_p$) from 1 to 10 and $\delta_p$ from 0.3 to 2 induces a progressive decrease in the phase shift between Cdc42-GTP and Pak1. This shift declines as the pair ($k_p$, $\delta_p$) reaches (10, 2), whereupon the oscillations are increasingly attenuated and ultimately extinguished. The separation between the dashed black lines illustrates the phase discrepancy between Cdc42-GTP and Pak1. **(C)** A comparison between model results with and without Pak1-dependent endocytosis.

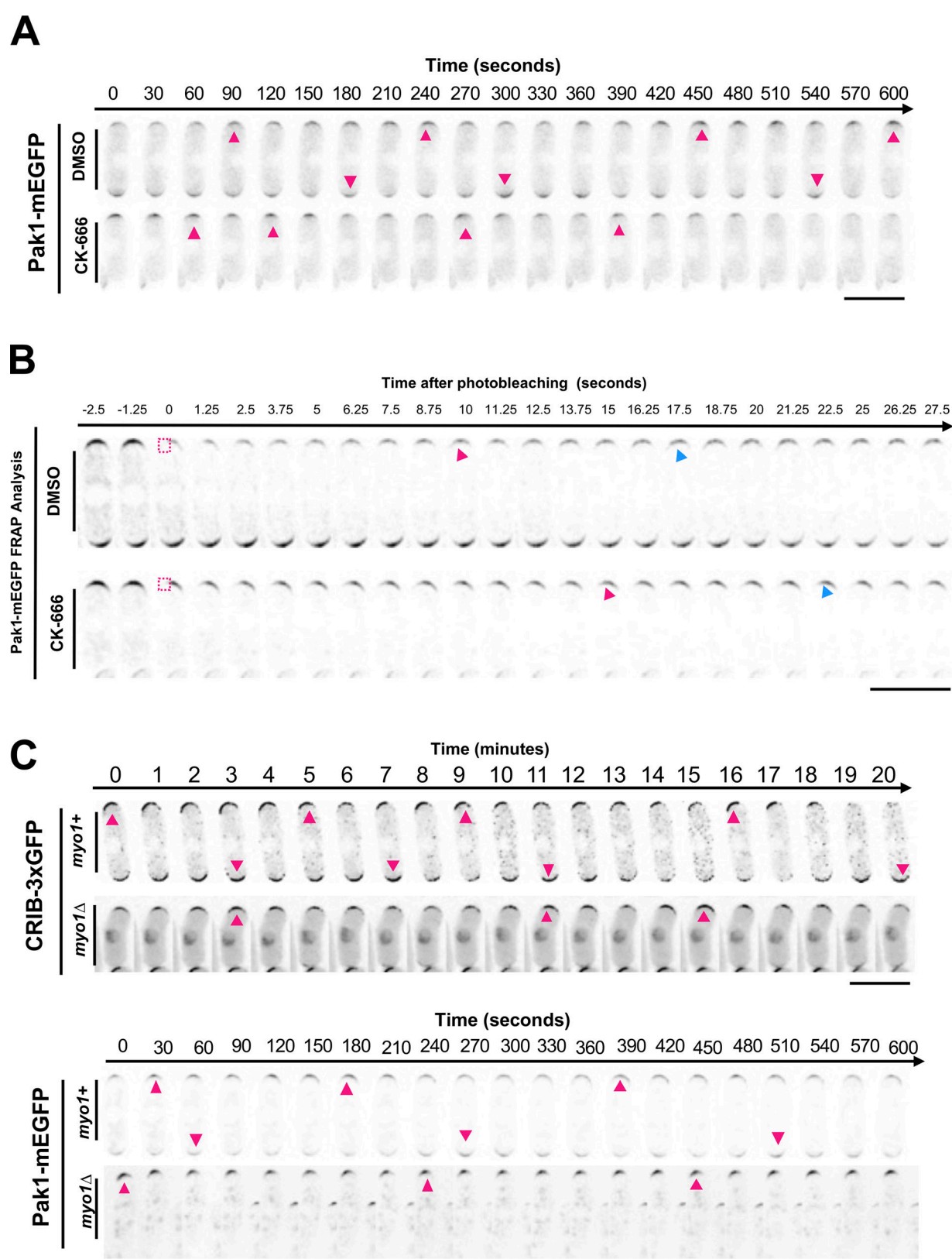

Figure S5. **Pak1-mEGFP and CRIB-GFP dynamics under different conditions. (A)** Pak1-mEGFP dynamics in DMSO and CK-666 treated cells. Magenta arrows indicate peaks of fluorescence intensity at that cell end. **(B)** FRAP analysis of Pak1-mEGFP recovery after photobleaching half the cell end. Magenta boxes indicate the half of the cell end that has been photobleached. Magenta arrows show when half of the final fluorescence has recovered. Blue arrows indicate when the fluorescence recovery plateaus. **(C)** Dynamics of CRIB-GFP and Pak1-mEGFP were observed in *myo1+* and *myo1Δ* cells. Magenta arrows indicate peaks of fluorescent intensity at that cell end. Oscillation montages clarified and denoised, made in NIS elements. Scale bars, 10 μm.

**Video 1.** **Active Cdc42 oscillations in a DMSO-treated *S. pombe* cell.** Active Cdc42 probe CRIB-3xGFP oscillates at the cell ends in an interphase cell treated with DMSO. Imaged with a laser scanning confocal microscopy system in 1-min intervals for 1 h. Displayed at 7 frames per second. Supplement for Fig. 1 A.

**Video 2.** **Active Cdc42 oscillations in a CK-666 treated *S. pombe* cell.** Oscillations at the cell ends of the active Cdc42 probe CRIB-3xGFP are disrupted in cells treated with 100 μm CK666. Imaged with a laser scanning confocal microscopy system in 1-min intervals for 1 h. Displayed at 7 frames per second. Supplement for Fig. 1 A.

**Video 3.** **Pak1 kinase oscillations in a DMSO-treated *S. pombe* cell.** Pak1-mEGFP displays oscillatory behavior at the cell ends in interphase cells. Imaged with time-lapse epifluorescence microscopy in 30-s intervals for 20 min. Displayed at 7 frames per second. Supplement for Fig. S5 A.

**Video 4.** **Pak1 kinase oscillations in a CK-666 treated *S. pombe* cell.** Pak1-mEGFP oscillatory behavior is disrupted in cells treated with 100 μm CK666. Pak1-mEGFP appears to stabilize at the ends upon CK666 treatment. Imaged with time-lapse epifluorescence microscopy in 30-s intervals for 20 min. Displayed at 7 frames per second. Supplement for Fig. S5 A.

**Provided online are Table S1, Table S2, Table S3, and Data S1. Table S1 shows parameter values of the wild-type model2, including parameters in model2-1. Table S2 shows parameter set comparisons. Table S3 shows the strain list. Data S1 shows data for the plots shown in the figures of the article.**

### Mathematical Modeling

**Model 1:** Based on the Xu-Jilkine model [1], this model employs reaction-diffusion equations to delineate the regulatory network, yet it is founded on networks that are distinct from those of the original. The network in the present model encompasses active molecules such as Cdc42-GTP and Scd1 at the cell tips, in addition to the diffusion of their inactive counterparts, Cdc42-GDP and Scd1, throughout the cytoplasm.

In the model, the total amounts of Cdc42($C_{total}$) and Scd1($S_{total}$) are conserved (constant), encompassing both cytoplasmic and tip-localized Cdc42 and Scd1. The two cell tips engage in competition for the acquisition of Cdc42 and Scd1. Here, $C(x,t)$ represents the inactive Cdc42 in the cytoplasm, and $S(x,t)$ denotes the cytoplasmic Scd1. The terms $c_i(t)$ and $s_i(t)$, where $i = 1, 2$, correspond to the active Cdc42 and the accumulated Scd1 at the tips, respectively. $L$ is the length of cell cytoplasm. The volume of each tip, $V_i$ (with $i = 1, 2$), is assumed to be one unit:

$$\sum_{i=1}^{2} c_i V_i + \int_0^L C(x,t)dx = C_{total} \quad (1)$$

$$\sum_{i=1}^{2} s_i V_i + \int_0^L S(x,t)dx = S_{total} \quad (2)$$

The diffusion part in the cytoplasm derived from Eq. 1 and Eq. 2 is:

$$\frac{\partial C}{\partial t}(x,t) = D_C \frac{\partial^2 C}{\partial x^2}(x,t) \quad (3)$$

$$\frac{\partial S}{\partial t}(x,t) = D_S \frac{\partial^2 S}{\partial x^2}(x,t) \quad (4)$$

$D_C$ and $D_S$ are diffusion coefficients of Cdc42-GDP and Scd1, respectively.

The reaction part at tips is

$$\frac{dc_i}{dt}(t) = k^c(s_i(t))C(L_i,t) - \delta_c c_i(t), i = 1, 2 \quad (5)$$

$$\frac{ds_i}{dt}(t) = k^s(c_i(t))S(L_i,t) - \delta_s s_i(t), i = 1, 2 \quad (6)$$

$$k^c(s_i(t)) = \frac{k_c s_i(t)^{n_{sc}}}{s_i(t)^{n_{sc}} + K_{sc}^{n_{sc}}} \quad (7)$$

$$k^s(c_i(t)) = \frac{k_s c_i(t)^{n_{cs}}}{\left(c_i(t)^{n_{cs}} + K_{cs}^{n_{cs}}\right)\left(1 + \left(\frac{c_i(t)}{K_{as}}\right)^{n_{as}}\right)} \quad (8)$$

$c_i$, $g_i$, and $p_i$ are active Cdc42, Scd1, and Pak1 at two tips. $\delta_C$ is the detachment rate constant for molecule Cdc42-GTP. $\delta_S$ is the detachment rate constant for molecule Scd1. One time unit is 1 min.

**Model 2:** The structural design of Model 2 closely resembles that of Model 1. The network includes Cdc42-GTP, protein X (Pak1), and Scd1 at the tips, as well as the diffusion of inactive molecules like Cdc42-GDP, protein X (Pak1), and Scd1 within the cytoplasm. Pak1 is described as protein X at first. In model 2, the detachment term for Pak1, denoted as X, is given by $\delta_x x_i(t)$. Following identification of X as Pak1, which can promote the endocytosis, the detachment term is updated to $\delta_p(k_o + f(p_i(t)))p_i(t)$. $k_o$ represents other elements in the cell which can also promote endocytosis

The diffusion part:

$$\frac{\partial C}{\partial t}(x,t) = D_C \frac{\partial^2 C}{\partial x^2}(x,t) \quad (9)$$

$$\frac{\partial S}{\partial t}(x,t) = D_S \frac{\partial^2 S}{\partial x^2}(x,t) \quad (10)$$

$$\frac{\partial P}{\partial t}(x,t) = D_P \frac{\partial^2 P}{\partial x^2}(x,t) \quad (11)$$

$D_C$, $D_S$, and $D_P$ are diffusion coefficients of Cdc42-GDP, Scd1, and Pak1, respectively.

The equations at tips for the second model are:

$$\frac{dc_i}{dt}(t) = k^c(s_i(t))C(L_i,t) - \delta_c c_i(t), i = 1, 2 \quad (12)$$

$$\frac{ds_i}{dt}(t) = k^s(c_i(t), p_i(t))S(L_i,t) - \delta_s s_i(t), i = 1, 2 \quad (13)$$

$$\frac{dp_i}{dt}(t) = k^p(c_i(t))P(L_i,t) - \delta_p(k_o + f(p_i(t)))p_i(t), i = 1, 2 \quad (14)$$

$$k^c(s_i(t)) = \frac{k_c s_i(t)^{n_{sc}}}{s_i(t)^{n_{sc}} + K_{sc}^{n_{sc}}} \quad (15)$$

$$k^s(c_i(t), p_i(t)) = \frac{k_s c_i(t)^{n_{cs}}}{\left(c_i(t)^{n_{cs}} + K_{cs}^{n_{cs}}\right)\left(1 + \left(\frac{p_i(t)}{K_{ps}}\right)^{n_{ps}}\right)} \quad (16)$$

$$k^p(c_i(t)) = \frac{k_p c_i(t)^{n_{cp}}}{c_i(t)^{n_{cp}} + K_{cp}^{n_{cp}}} \quad (17)$$

$$f(p_i(t)) = \frac{p_i(t)^{n_{pe}}}{p_i(t)^{n_{pe}} + K_{pe}^{n_{pe}}} \quad (18)$$

$c_i$, $g_i$, and $p_i$ are active Cdc42, Scd1, and Pak1 at two tips. $\delta_C$ and $\delta_S$ are the same in Model 1; $\delta_p$ is the detachment rate constant for molecule Pak1. f(p) describes that Pak1 promotes endocytosis.

The inclusion of activation of endocytosis by Pak1 locally (Eq. 18) does not change the general behaviors of the model significantly, e.g., the bipolarity and anti-correlated oscillation. Nonetheless, comparing two versions of Model 2 (without Pak1-dependent endocytosis regulation: $f(p_i(t)) \equiv 1$, and with Pak1-dependent endocytosis regulation: Eq. 18), we observed some minor differences, including larger amplitude of Pak1 oscillation, and smaller amplitudes of Cdc42-GTP and Scd1 oscillations with the inclusion of Pak1-dependent endocytosis regulation (Fig. S4 C).

**Model 3:** Incorporating another negative feedback loop into the Cdc42-GTP regulatory network, we seek to provide an explanation for bipolarity in the absence of Pak1. However, the specific details of this second feedback loop are unknown. As a result, we made the assumption that active Cdc42 can inhibit itself, either directly or indirectly. Eq. 12 is changed to:

$$\frac{dc_i}{dt}(t) = k^c(s_i(t), c_i(t))C(L_i,t) - \delta_c c_i(t), i = 1, 2 \quad (12a)$$

and Eq. 15 is changed to:

$$k^c\big(s_i(t), c_i(t)\big) = \frac{k_c s_i(t)^{n_{sc}}}{s_i(t)^{n_{sc}} + K_{sc}^{n_{sc}}} e^{\frac{-c_i(t)}{a1}} \quad (15a)$$

Here $a1$ is a constant describing the strength of the second negative feedback loop.

Comparison of oscillatory dynamics between Model 1 and Model 2: In MATLAB, Sobol parameter spaces are defined for each model, with each model having 100,000 sample points in its parameter space. The objective is to detect oscillatory dynamics by counting peaks and measuring the differences in amplitude between neighboring peaks. The parameter sets that lead to oscillatory dynamics are counted and used to calculate the percentage of occurrences of oscillatory dynamics. (Range of parameters are in Table S1.)

**MATLAB CODE:**

```
%Below is code for WT, CK666 is decreasing detachment of Pak1
    time = 1000; tspan = [0:0.1:time]; tsteps = time/0.01; hx = 0.1;
    L = 1;
    xspan = 0:hx:L; n = numel(xspan); %initial condition
    C0 = 0.15*ones(n,1); G0 = 0.1*ones(n,1);
    P0 = 0.1*ones(n,1);
    cgp = [1.3;0.2;1.07;0.2;1.3;0.2];
    u0 = [C0;G0;P0;cgp]; % parameters
    Dc = 3;
    Dg = 3;
    Dp = 3;
    k0 = 1.7;
    nsc = 5; ksc = 0.5; deltac = 0.3;
    k_on = 1.5;
    ncs = 1;
    kcs = 1;
    kps = 0.1; nps = 3; deltas = 0.1; kpa = 1; ncp = 3; kcp = 1; deltap = 0.3; npe = 3; kpe = 0.3; p = [Dc;Dg;k0;nsc;ksc;deltac; k_on;ncs;kcs;
kps;nps;deltas;kpa;ncp;kcp;deltap;Dp;as;npe;kpe]; % solve
    [T ,Y] = ode15 s(@(t,u) ode_solve(t,u,p,n,hx),tspan,u0);
    y1 = real(round(Y(:,end-5),4));% round a decimal to 4 digits
    y11 = real(round(Y(:,end-4),4));
    y2 = real(round(Y(:,end-3),4));
    y22 = real(round(Y(:,end-2),4));
    y3 = real(round(Y(:,end-1),4));
    y33 = real(round(Y(:,end),4));
    figure;
    set(gcf, 'Position', [100, 100, 600, 270]);
    % Plot only the first graph
    ax1 = axes; % Use axes for a single plot
    plot(ax1,T,y1,'color',[0.1 0.7 0.4],'linewidth',2.5); hold on; plot(ax1,T,y2,'color',[0.1010 0.6 1],'linewidth',2.5); hold on;
    %}
    plot(ax1,T,y3,'color',[0.9 0.2 0.1],'linewidth',2.5);
    % Set the thickness of the plot boundary
    set(ax1, 'LineWidth', 2); % Increase this value for a thicker boundary
    % Add title
    title(['Model of Protein Dynamics at One Cell End, k_{p} = ' num2str(kpa) ', \deltap = ' num2str(deltap)],'FontSize',18);
    % Adjust legend
    h = legend('Cdc42-GTP', 'Scd1', 'Pak1', 'Location', 'Best', 'Orientation', 'horizontal', 'Box', 'off');
    set(h, 'FontSize', 15);
    set(gca, 'FontSize', 12);
    % Adjust title and labels
    xlabel('Time (a.u.)', 'FontSize', 17); ylabel('Intensity', 'FontSize', 17); xlim([100 200]); ylim([0 1.5])
    % Save the figure as SVG
    print(gcf,'myPlot.jpg','-djpeg','-r600'); function F = ode_solve(t,u,p,n,hx)
    c1 = u(3*n+1);
    c2 = u(3*n+2);
```

```matlab
    g1 = u(3*n+3);
    g2 = u(3*n+4);
    pa1 = u(3*n+5);
    pa2 = u(3*n+6);
    Dc = p(1);
    Dg = p(2);
    k0 = p(3);
    nsc = p(4);
    ksc = p(5);
    deltac = p(6);
    k_on = p(7);
    ncs = p(8);
    kcs = p(9);
    kps = p(10);
    nps = p(11);
    deltas = p(12);
    kpa = p(13);
    ncp = p(14);
    kcp = p(15);
    deltap = p(16);
    Dp = p(17);
    as = p(18);
    npe = p(19); kpe = p(20); kplus = @(x) k0*x^nsc/(x^nsc+ksc^nsc); kon = @(y,z) k_on*(y^ncs+0)/(kcs^ncs+y^ncs).*1/(1+as*(z/kps)
^nps); kp = @(y) kpa*y^ncp/(kcp^ncp+y^ncp); ep = @(z) (0.2+(z^npe)/(z^npe+kpe^npe)); rc = Dc/(hx^2); rg = Dg/(hx^2); rp = Dp/
(hx^2);
    F(1) = rc*(-(2+2*hx*kplus(g1)/Dc).*u(1) + 2*u(2) + 2*hx*deltac*c1/Dc); %DCDt(1)
    F(2:n-1) = rc*(u(1:n-2) -2*u(2:n-1) + u(3:n));% DCDt(2:n-1)
    F(n) = rc*(2*u(n-1) - (2+2*hx*kplus(g2)/Dc).*u(n) + 2*hx*deltac*c2/Dc); F(n+1) = rg*(-(2+2*hx*kon(c1,pa1)/Dg).*u(n+1) + 2*u(n+2)
+ 2*hx*deltas*g1/Dg); %DGDt
    F(n+2:2*n-1) = rg*(u(n+1:2*n-2) -2*u(n+2:2*n-1) + u(n+3:2*n));
    F(2*n) = rg*(2*u(2*n-1) - (2+2*hx*kon(c2,pa2)/Dg).*u(2*n) + 2*hx*deltas*g2/Dg); F(2*n+1) = rp*(-(2+2*hx*kp(c1)/Dp).*u(2*n+1) +
2*u(2*n+2) + 2*hx*deltap*pa1/Dp); %DGDt
    F(2*n+2:3*n-1) = rp*(u(2*n+1:3*n-2) -2*u(2*n+2:3*n-1) + u(2*n+3:3*n));
    F(3*n) = rp*(2*u(3*n-1) - (2+2*hx*kp(c2)/Dp).*u(3*n) + 2*hx*deltap*pa2/Dp);
    F(3*n+1) = kplus(g1).*u(1) - deltac*c1;
    F(3*n+2) = kplus(g2).*u(n) - deltac*c2;
    F(3*n+3) = kon(c1,pa1).*u(n+1) - deltas*g1;
    F(3*n+4) = kon(c2,pa2).*u(2*n) - deltas*g2;
    F(3*n+5) = kp(c1).*u(2*n+1)-deltap*ep(pa1)*pa1;
    F(3*n+6) = kp(c2).*u(3*n)-deltap*ep(pa2)*pa2;
    F = F(:);
    end
```

