## [Peer Review File · The Journal of Cell Biology]

Arp2/3-dependent endocytosis ensures Cdc42 oscillations by removing Pak1-mediated negative feedback

Marcus Harrell, Ziyi Liu, Bethany Campbell, Olivia Chinsen, Tian Hong, and Maitreyi Das

Corresponding Author(s): Maitreyi Das, Boston College

Review Timeline:

Submission Date:	2023-11-21
Editorial Decision:	2024-01-11
Revision Received:	2024-05-10
Editorial Decision:	2024-05-22
Revision Received:	2024-06-21

Monitoring Editor: Daniel Lew

Scientific Editor: Tim Fessenden

Transaction Report:

DOI: <https://doi.org/10.1083/jcb.202311139>

January 11, 2024

Re: JCB manuscript #202311139

Dr. Maitreyi Das
Boston College
Biology
140 Commonwealth Avenue
Chestnut Hill, MA 02467

Dear Dr. Das,

Your manuscript has now been reviewed by three referees whose comments are appended below. As you will see, all are intrigued by the novel ideas presented, including the idea that Pak1 is removed from the cortex by endocytosis and the idea that endocytic removal of Pak1 is important to enable the oscillation of Cdc42 between the poles. However, while your findings on the effects of Arp2/3 inhibition are consistent with those ideas, the reviewers were not entirely convinced regarding your proposed mechanisms. We invite you to revise your paper in response to the reviewer comments, which we summarize in points 1 and 2 below. Additional data sought by reviewers besides these two issues are not required in a revision. I too am very intrigued by your work and had some additional thoughts in points 3-6 that might help explain why reviewers (and I) were confused on mechanistic issues.

1. Is Pak1 removed from the membrane by endocytosis? Both reviewer 2 and 3 were skeptical about this, and indicated that you need to distinguish whether Pak1 is on the endocytic vesicle or is perhaps indirectly affected by the act of endocytosis. I too am confused as to exactly what you envision to happen. If molecules of Pak1 that are on the membrane at a patch get endocytosed, I would not expect the concentration of Pak1 at the remaining plasma membrane to change (since endocytosis would remove both membrane and the associated Pak1, leaving the neighboring concentration untouched). Why would this create a dark "hole" in the Pak1 signal in the remaining (non-endocytosed) membrane? This is a rather different case than the endocytic proteins that you make an analogy to. I would also point out that nobody has reported such "dark holes" for cargos like v-SNAREs that are universally agreed to be removed by endocytosis.

2. Does Cdc42 or Pak1 promote endocytosis? Reviewer 2 did a deep dive on several of your claims here and did not find much supporting evidence in the literature you cited. This needs to be corrected/clarified. My recollection of the old literature in *S. cerevisiae* is that Cla4 can phosphorylate type I myosin, which was thought at the time to regulate polarity, not endocytosis. And the physiological significance of early claims about Cdc42 and PAKs promoting actin patch polymerization in permeabilized *S. cerevisiae* cells was later questioned by findings that mutants lacking Cdc42 or PAKs continued to actively endocytose as they grew larger. If you know of specific experiments that show a role for Cdc42 or PAKs in endocytosis, please explain what those are and cite the relevant papers. If not (which is fine), then your findings on this are actually more novel than you had thought!

3. What is the entity removed by endocytosis? A related point that had me confused is whether Pak1 alone or a GTP-Cdc42-Pak1 complex gets endocytosed. All studies I am aware of have posited that PAK localization depends on on-going interaction with membrane-associated GTP-Cdc42. Is that your view? If so, this may merit revision of the text and model to clarify.

4. What are the assumptions in your model with regard to point 2 above? Your models currently have independent "degradation" terms (I would suggest that "detachment" would be a better term than "degradation" for this reaction) for Cdc42 and Pak1. This would seem to say that Pak1 can detach from Cdc42 and yet remain attached to the membrane until it is removed by endocytosis. Is that your view? Related to this, in your modeling time-series it would appear that Pak1 detaches from the membrane with a significant delay after Cdc42 has detached (e.g. Fig. 7A). If I read this correctly, this delayed detachment is responsible for the "offset" you analyze in Fig. 7. Would this disappear if the model coupled the detachment of Pak1 and Cdc42?

5. Is independent detachment of Pak1 and Cdc42 critical for your conclusions? If so, it may be helpful to show some evidence for this key assumption. For example, FRAP analysis of Pak1 vs Scd1/Scd2 or Cdc42 could demonstrate that Pak1 is more stable (detaches more slowly). FRAP of Pak1 in control versus CK666 treatment could also provide important support for your assumption that Pak1 is less dynamic when endocytosis is inhibited.

6. Does Pak1 regulate endocytosis globally or locally? You say (lines 349-350) that "Pak1 kinase has been shown to phosphorylate Myo1 *during endocytosis*" (emphasis mine) but you do not cite any paper for that and I am not aware of any evidence that phosphorylation occurs "in the act". Similarly, your model 2.1 assumes that local endocytosis rate is a Hill function of the local Pak1 concentration. But so far as I can tell, it could be that global Pak1 activity is required to maintain Myo1 phosphorylation for effective endocytosis. How important is it for your conclusions that Pak1 regulation occurs globally vs locally? If the model works equally well with global regulation, then perhaps you could change the model to be simpler. If, however, your conclusions are dependent on the steep local regulation currently assumed, then you might consider providing some evidence for local control.

GENERAL GUIDELINES:

Text limits: Character count for an Article is < 40,000, not including spaces. Count includes title page, abstract, introduction, results, discussion, and acknowledgments. Count does not include materials and methods, figure legends, references, tables, or supplemental legends.

Figures: Articles may have up to 10 main text figures. Figures must be prepared according to the policies outlined in our Instructions to Authors, under Data Presentation, <https://jcb.rupress.org/site/misc/ifora.xhtml>. All figures in accepted manuscripts will be screened prior to publication.

*****IMPORTANT:** It is JCB policy that if requested, original data images must be made available. Failure to provide original images upon request will result in unavoidable delays in publication. Please ensure that you have access to all original microscopy and blot data images before submitting your revision. ***

Supplemental information: There are strict limits on the allowable amount of supplemental data. Articles may have up to 5 supplemental figures. Up to 10 supplemental videos or flash animations are allowed. A summary of all supplemental material should appear at the end of the Materials and methods section.

Please note that JCB now requires authors to submit Source Data used to generate figures containing gels and Western blots with all revised manuscripts. This Source Data consists of fully uncropped and unprocessed images for each gel/blot displayed in the main and supplemental figures. Since your paper includes cropped gel and/or blot images, please be sure to provide one Source Data file for each figure that contains gels and/or blots along with your revised manuscript files. File names for Source Data figures should be alphanumeric without any spaces or special characters (i.e., SourceDataF#, where F# refers to the associated main figure number or SourceDataFS# for those associated with Supplementary figures). The lanes of the gels/blots should be labeled as they are in the associated figure, the place where cropping was applied should be marked (with a box), and molecular weight/size standards should be labeled wherever possible.

The typical timeframe for revisions is three to four months. While most universities and institutes have reopened labs and allowed researchers to begin working at nearly pre-pandemic levels, we at JCB realize that the lingering effects of the COVID-19 pandemic may still be impacting some aspects of your work, including the acquisition of equipment and reagents. Therefore, if you anticipate any difficulties in meeting this aforementioned revision time limit, please contact us and we can work with you to find an appropriate time frame for resubmission. Please note that papers are generally considered through only one revision cycle, so any revised manuscript will likely be either accepted or rejected.

Thank you for this interesting contribution to Journal of Cell Biology. You can contact us at the journal office with any questions at cellbio@rockefeller.edu.

Sincerely,

Daniel Lew
Monitoring Editor
Journal of Cell Biology

Tim Fessenden
Scientific Editor
Journal of Cell Biology

Reviewer #1 (Comments to the Authors (Required)):

Endocytosis is known to modulate cell growth and shape by affecting cell polarization and exocytosis, but the underlying mechanisms are not clearly understood. In the rod-shaped fission yeast, the Cdc42-GTP (indicated by CRIB-3xGFP) displays anti-correlated oscillations between the cell ends, which is essential for bipolar growth in this organism. This oscillatory behavior requires a positive feedback mechanism involving Scd1 (GEF for Cdc42) and Scd2 (scaffold) and a time-delayed negative feedback mechanism involving Pak1 (an effector of Cdc42). In this study, the authors report that disruption of endocytosis by the Arp2/3-specific inhibitor CK-666 disrupts CRIB oscillation and disables the positive feedback mechanism (by decreasing the localization of Scd1 and Scd2 at the cell ends). This is an important finding that is well supported by the data. Through further experimentation and modeling, the authors conclude that the Cdc42-dependent recruitment of Pak1 to a cell end, when reached to a critical level, will disable the positive feedback mechanism presumably phosphorylating Scd1 and Scd2, causing Scd1-Scd2 dissociation (as suggested by previous studies), and that endocytic removal of Pak1 is the key for the return of Cdc42 activation to the cell end. Furthermore, the authors conclude that Pak1 may stimulate its own removal by facilitating endocytosis. While this Pak1-centered idea is quite attractive, further experiments and clarification are required to establish its validity.

Major points:

1. To test the role of endocytic removal of Pak1 in Cdc42 oscillations as well as the role of Pak1 in endocytosis, the same of panel of oscillation and endocytosis makers (i.e., CRIB, Scd1, and Fim1) should be examined by time-lapse analysis in cells [such as GFP-pak1-as (analog sensitive) cells] where Pak1 localization can be monitored, and its activity can be acutely inactivated. These data should be compared to the data derived from CK-666-treated cells to determine their similarities and differences. This will help determine to what degree the impact of endocytosis on Cdc42 oscillation is mediated by Pak1 activity, to what degree Pak1 controls endocytosis, and to what degree Pak1 is removed by endocytosis. If the pak1-as is not available, the same experiments should be performed using the pak1-Ts strain (not simply monitoring the Scd1 intensity as done in Fig. 4B).
2. CK-666 treatment appears to mainly dampen the dynamics of CRIB (Fig. 1A and Fig. S1B, right) and Pak1 (Fig. 3C, right) at one end, likely the old end (Fig. S1B, right), while the oscillations (or on/off) of CRIB and Pak1 at the other end appear normal. Why and how?
3. The impact of Myo1 deletion on Scd1 and Pak1 dynamics (Fig. 5) seems to be stronger than that caused by CK-666 treatment, yet the effect of Myo1 deletion on CRIB dynamics appears to be weaker than that by CK-666 treatment. Why?

Minor points:

1. Fig. 1, A and B: while anti-correlated oscillations are maintained in for3 cells, the CRIBX3GFP signal at the cell ends appear much stronger than that in WT cells. If so, how?
2. Line 235, change "Fig. 3B" to "Fig. 2B"
3. Line 301, Change "Fig. 5A, B" to "Fig. 4A, B"
4. Discuss more about the similarities and differences between the Scd1-Scd2-pak1 in fission yeast and the Cdc24-Bem1-Cla4 in budding yeast in terms of their roles in controlling the dynamics of Cdc42 polarization. The paper on the role of Cla4 in regulating polarized growth by phosphorylating Cdc24 (Gulli et al., Mol. Cell. 2000 by Mathias Peter's group) and the paper on the dispersal of Cdc42 by endocytosis (Irazoqui et al., MBoC. 2005 by Daniel Lew's group) should be included for discussion.

Reviewer #2 (Comments to the Authors (Required)):

This study by Harrell et al focuses on a potential mechanism of removal of Pak1 kinase-mediated negative feedback, which is essential for Cdc42 oscillations between two cell ends that are responsible for polarized bipolar growth of fission yeast *S. pombe* cells. The authors discovered that blocking endocytosis through treatment with Arp2/3 complex inhibitor CK-666 or the use of myo1delta mutant (which is viable but has severe endocytosis defect) stops Cdc42 and Pak1 oscillations, decreases accumulation of Cdc42 GEF Scd1, and increases accumulation of Pak1. Through mathematical modeling and excellent 2-color live cell imaging the authors provide strong evidence that Pak1 is responsible for delayed negative feedback, which is essential for Cdc42 oscillations. For this model to work, Pak1 needs to be eventually removed and, based on their CK-666 and myo1delta data the authors propose that Arp2/3 complex-mediated branched actin-dependent endocytosis is directly responsible for Pak1 removal. Indeed, when the authors looked at Pak1 at endocytic sites, they observed that Pak1 briefly disappeared from the membrane at the moment an endocytic actin patch internalized. However, the authors did not see Pak1 internalizing with the patch, which is very surprising and casts doubt on the authors' overarching conclusion that endocytosis is directly responsible for Pak1 removal. This is a major concern and below I propose additional experimentation that may verify or defy this conclusion.

Overall, the data in this manuscript are of good quality but I wish I could say the same about the writing and especially citations, as detailed below. The manuscript in its current form is not acceptable for publication in JCB and needs major revision as well as a little bit of additional experimentation.

Major concerns:

(1) The observations that blocking endocytosis with CK-666 or myo1delta results in increased accumulation of Pak1 is very convincing and extremely intriguing. It does suggest that actin-mediated endocytosis does play some important role in Pak1 removal. The authors' observation that Pak1 level on the membrane at the endocytic site drop at the exact moment of actin patch internalization is equally convincing but the fact that the authors do not see Pak1 internalizing with the patch suggests that endocytosis does not remove Pak1 directly (i.e. removing Pak1 that is associated with the membrane of an endocytic vesicle) but does so indirectly through some thus far unknown mechanism. The authors can address this concern both experimentally and by changing the wording throughout the ms:

(a) Would it make sense to go back and take a look with a very sensitive detector to make absolutely sure that no Pak1 is internalizing with the patch? In doing so, the authors would need to be very careful to include Fim1-mCherry only control for bleed through.

(b) The authors have an excellent opportunity to test their conclusions by repeating Pak1-EGFP and Fim1-mCherry 2-color imaging in myo1delta cells. Over half of actin patches in myo1delta fail to internalize. So, if the authors are right, they should see that in myo1delta cells Pak1 does not disappear from the membrane when patches do not internalize. If Pak1 continues to disappear in non-internalizing patches, that would suggest that some other mechanisms rather than direct endocytosis are at play.

(c) Throughout the ms, including the title, the authors may want to consider softening the language by replacing statements like "endocytosis removes Pak1" with something less direct like "endocytosis plays a role in, or is important for Pak1 removal". Likewise, it would be better to re-word the title into something like "Arp2/3- and Myo1-dependent removal of Pak1 facilitates..." Also, please consider softening the title of Figure 6 "Pak1 is removed from the cell end via endocytosis".

(d) In the Discussion, the authors did not offer any explanations or potential mechanisms for their most intriguing observation as to how Pak1 is removed from the membrane at the endocytic sites without being internalized with the endocytic vesicles.

(2) There are serious problems with citations throughout the manuscript. The authors, like this reviewer, should invest some time into double- and triple-checking their references to make sure that the references appropriately support their statements. Here are just five (out of many) examples:

(a) Introduction, p7: Active Cdc42 also triggers branched actin polymerization, necessary for endocytosis in yeasts, through the WASP protein, Wsp1, and the type 1 myosin, Myo1 (Bendezú and Martin, 2011; Gachet and Hyams, 2005; Kolluri et al., 1996; Kovar et al., 2011; Landino et al., 2021; Lechler et al., 2000; Lee et al., 2000; Sirotkin et al., 2005).

First of all, the statement that "Active Cdc42 also triggers branched actin polymerization ... through the WASP protein, Wsp1..." is factually not true for yeast. Budding yeast WASp Las17 lacks GTPase binding domain (GBD) and it is not clear if *S. pombe* Wsp1 has a GBD and as far as know, nobody looked at it in *S. pombe* experimentally. Even in mammalian cells, while it is true that WASp and N-WASp are activated by Cdc42, it is still not clear if this mode of activation occurs at the sites of endocytosis. Now to the references. Bendezú and Martin, 2011, Gachet and Hyams, 2005, Landino et al., 2021, Lee et al., 2000, Sirotkin et al., 2005 make no mention of Cdc42 activating Wsp1. Kovar et al., 2011 only mention Cdc42 in the context of actin cables. Lechler et al., 2000 (as well as Lechler et al., 2001, which authors failed to cite) is a budding yeast work that talks about Cdc42-Pak1-myosin 1 activation but not Wsp1. Kolluri et al. is a mammalian WASp reference. Thus, none of the papers cited by the authors support their statement.

(b) Discussion, p29: Likewise, Cdc42 activation also promotes actin polymerization for endocytosis and exocytosis (Adamo et al., 2001; Campbell et al., 2022; Estravis et al., 2012; Estravis et al., 2017; Estravis et al., 2011; Gachet and Hyams, 2005; Kovar et al., 2011; Lamas et al., 2020a; Sagot et al., 2002).

Every single one of these 9 references deals with the role of Cdc42 in exocytosis only. No references are provided for the role of Cdc42 in endocytosis.

(c) Discussion, p26: Bipolarity is established when the new end is able to overcome the old end dominance and robustly activate Cdc42 (Mitchison and Nurse, 1985).

Unfortunately, this classic paper makes no mention of Cdc42.

(d) Results, p21: While the role for Pak1 kinase in endocytosis has not been fully investigated, it has been shown to phosphorylate Myo1 during endocytosis (Attanapola et al., 2009; Lee et al., 2000).

Lee et al., 2000 have not shown that. They only mentioned a potential role of Pak1 in phosphorylating Myo1 in the Discussion, based on prior budding yeast work, which authors of the current ms neglected to cite.

(e) Discussion, p24: Arp2/3 dependent endocytosis is known to be required for growth in *S. pombe* and other fungal species as well as mammalian cells with high membrane tension (Aghamohammadzadeh and Ayscough, 2009; Basu et al., 2014b; Epp et al., 2010).

All three of these are yeast references. The authors must cite Boulant et al (2011) Nature Cell Biology for mammalian cells.

(3) The ms is rather on the verbose side and can greatly benefit from some serious cutting and editing. This is especially evident in the Introduction. Additionally, several sections of the Discussion are redundant with the Introduction, for example, the entire first paragraph of page 26.

Minor concerns:

Results, p21. Unfortunately, the authors switched from correctly calling Fim1-mCherry patches endocytic actin patches (or simply actin patches or patches) to calling them endocytic vesicles. This is unfortunate because they made this switch while describing endocytic defects in pak1-ts mutant, where actin patches marked by Fim1-mCherry still form but the endocytic vesicles may fail to form, invaginate or internalize. I suggest the authors change the wording on this page from "endocytic vesicles" to "actin patches".

The authors may want to consider moving their excellent Fig. 7 to earlier in the paper, for example right after Fig. 2, as the data in current Fig. 7 strongly support the role of Pak1 as being responsible for the delayed negative feedback.

Introduction, p7: The statement that "While Cdc42 has been shown to regulate endocytosis and actin organization" needs to be supported by appropriate references if such references exist, especially as it pertains to endocytosis.

Discussion, p27: "... removal of negative feedback by Pak1-dependent endocytosis..."
Do you mean "removal of Pak1-dependent negative feedback by endocytosis"?

Figure 2B. Would it make sense to add labels Model 2, Cytoplasm, Cell Ends directly to this panel rather than rely on the Figure Legend alone?

Figure 3B vs 3D. The Pak1-mEGFP signal in the CK-666 treated image in Fig. 3B looks dimmer than in DMSO control image, contrary to quantitation in Fig. 3D. Were the two panels in Fig. 3B equally adjusted or is there an error?

Figure 4C. Would it make sense to add an asterisk or some other label to the green negative error to highlight that this is a new addition unique to Model 3?

There are many typos or missing spaces scattered throughout the ms.

Reviewer #3 (Comments to the Authors (Required)):

This manuscript by Harrell, Liu et al., uncovers a regulatory link between clathrin-mediated endocytosis and the oscillations of active Cdc42 at sites of polarized growth in fission yeast. This is a very interesting paper and has important implications in understanding the coupling of membrane trafficking and polar growth. These mechanisms are likely widely conserved in many branches of eukaryotic life.

The experiments are carefully done and clearly presented. The conclusions are well supported by the provided genetic and cell biological data. I cannot comment on the technical aspects of the mathematical models, but they are well connected to the biological experiments and give a strong quantitative framework for thinking about the different possible regulatory networks. The manuscript is clearly written and logically structured. I believe it would be of great interest for the readership of JCB.

I have a few suggestions for further experiments. These are not experiments that I consider essential for the current manuscript, but they might add value to the work.

The exact mechanism by which Arp2/3 complex has an effect on Pak1 remains open. Is the effect mediated directly by the assembly of actin patches or does it require the formation of endocytic vesicles? The experiments with myosin mutants could shed some light on this question, but the authors do not describe in detail what is the phenotypic effect of the myo1 deletion on endocytic progression. Has this been described in previous studies in *S. pombe*? Knowing the effect of myosin deletion on endocytosis could help to understand the mechanism by which Arp2/3 activity regulates Pak1. The authors could do similar analyses on the myo1 mutant as they did in fig6 to characterize the pak1-ts phenotype.

Another endocytic mutant that could be informative in this context is the deletion of end4. This mutant blocks endocytosis, but allows robust Arp2/3 activation and actin assembly. However, the strong endocytosis block in the end4 mutants might make the experiments difficult.

Textual points:

- To me the title of the manuscript seems to imply that the authors are showing a direct connection between the Arp2/3 complex and Pak1, which obviously is not the case. I would rethink the title so that it better reflects the conclusions of the paper. For example: "Arp2/3 complex -dependent endocytosis periodically removes....".
- The authors state that Cdc42 regulates polarized growth in most eukaryotes. (lines 17, 48, 64). Maybe this is common knowledge in the polarization field, but could the authors cite the work that demonstrates this.
- On line 360 the authors cite Nakano et al., 2001. I don't think that this paper showed that "fimbrin internalizes into the cytoplasm with the endocytic vesicle".
- On lines 392 and 393 the authors state: Thus, Pak1 does not influence the ability of endocytic vesicles to form." Then on line 398 and 399 they write: "...indicating a greater fraction of failed endocytic events...". Please, clarify the text so that it doesn't sound contradictory.
- In the discussion section I was missing some discussion about the mechanism by which endocytosis removes Pak1. Uncovering this mechanisms is clearly beyond the scope of this manuscript, but it would be good to highlight this as an important open question and maybe propose some possibilities. There was actually some discussion about this point in the results section. Maybe that text would work better in the discussion section?

Draft Response to Reviewers

We thank the Editor and the Reviewers for their considered feedback on the manuscript. We have taken these suggestions and applied them to the current version of the manuscript. These suggested changes have further improved the manuscript and increased its clarity.

Response to the Editor,

1. Is Pak1 removed from the membrane by endocytosis? Both reviewer 2 and 3 were skeptical about this, and indicated that you need to distinguish whether Pak1 is on the endocytic vesicle or is perhaps indirectly affected by the act of endocytosis. I too am confused as to exactly what you envision to happen. If molecules of Pak1 that are on the membrane at a patch get endocytosed, I would not expect the concentration of Pak1 at the remaining plasma membrane to change (since endocytosis would remove both membrane and the associated Pak1, leaving the neighboring concentration untouched). Why would this create a dark "hole" in the Pak1 signal in the remaining (non-endocytosed) membrane? This is a rather different case than the endocytic proteins that you make an analogy to. I would also point out that nobody has reported such "dark holes" for cargos like v-SNAREs that are universally agreed to be removed by endocytosis.

Response: We agree that if Pak1 localizes all over the membrane at the cell ends, we should not see loss of Pak1 from the membrane during endocytosis. To test this, we carefully analyzed Pak1-mEGFP localization using deconvolution of 3-D reconstructed images of the cell ends. We also observed similar results using a spinning disc confocal microscope and the Zeiss Airyscan microscope. For the sake of clarity, we have shown the data with deconvolution. Our observations show that Pak1 at the cell ends do not exist as a cap but rather exists as distinct punctae. Moreover, we now show that these puncta of Pak1-mEGFP colocalize with Fim1-mCherry patches on the membrane

(Fig.8A) (p.21 lines 406-413). We hypothesize that these local concentrated areas of Pak1 are dissipating as endocytic patches internalize. At this time, photobleaching prevents us from performing 1-second interval time lapse imaging while also acquiring the number of Z-slices required to observe these Pak1 puncta from a bull's eye view as they are being lost. We have revised figure 8 to show the punctate localization pattern of Pak1-mEGFP at cell ends.

In that vein, we do not currently have the ability to distinguish if Pak1 itself is on endocytic vesicles or indirectly lost as a result of endocytosis. We can only conclusively say that we do not see Pak1 on internalized vesicles. It is possible that Pak1 would not stay associated with endocytic vesicles because active Cdc42 is not internalized, thus, there is no active Cdc42 to continue the recruitment of Pak1.

It was reported in budding yeast that type I myosin, myo1, does not internalize with the endocytic patch (Sirotkin et al 2005) while immuno-electron microscopy was able to detect a small portion (Idrissi et al 2008). It is possible that Pak1 is lost from the endocytic site in a manner similar to Myo1. Further analysis of Pak1 dynamics at the patch would explain this.

2. Does Cdc42 or Pak1 promote endocytosis? Reviewer 2 did a deep dive on several of your claims here and did not find much supporting evidence in the literature you cited. This needs to be corrected/clarified. My recollection of the old literature in S. cerevisiae is that Cla4 can phosphorylate type I myosin, which was thought at the time to regulate polarity, not endocytosis. And the physiological significance of early claims about Cdc42 and PAKs promoting actin patch polymerization in permeabilized S. cerevisiae cells was later questioned by findings that mutants lacking Cdc42 or PAKs continued to actively endocytose as they grew larger. If you know of specific experiments that show a role for Cdc42 or PAKs in endocytosis, please explain what those are and cite the relevant papers. If not (which is fine), then your findings on this are actually more novel than you had thought!

Response: We agree that these findings are more novel than we have indicated thus far. Reports by Murray and Johnson, 2001 as well as from our lab (Onwubiko et al, 2019, Campbell et al, 2022) have shown genetic evidence that Cdc42 regulates endocytosis. Pak1 kinase has also been reported as a potential kinase for Myo1 in fission yeast, however this study did not show conclusive biochemical evidence for this activity (Attanapola et al 2009). Current research in our lab is investigating how Cdc42 or Pak1 promotes endocytosis, but this data is part of a different manuscript and thesis. For now, we have revised our text to emphasize the novelty of these findings while also clarifying that future work will further address these findings (p.26 lines 484-491).

3. What is the entity removed by endocytosis? A related point that had me confused is whether Pak1 alone or a GTP-Cdc42-Pak1 complex gets endocytosed. All studies I am aware of have posited that PAK localization depends on on-going interaction with membrane-associated GTP-Cdc42. Is that your view? If so, this may merit revision of the text and model to clarify.

Response: We apologize for the lack of clarity in our previous explanation. Our data does not show Pak1 being endocytosed, instead we observe Pak1 loss at the site of endocytosis corresponding to when the vesicle starts to internalize. As such our data cannot demonstrate if Pak1 is bound to active Cdc42 when it is being lost. Indeed, active Cdc42 localization pattern differs from that of Pak1. While Pak1 is punctate, active Cdc42 at the cell ends shows both a diffused pattern as well as punctate. This diffused pattern prevents us from distinguishing whether Cdc42 is lost from the membrane along with Pak1 when the patch starts to internalize. However, we do not see active-Cdc42 (CRIB-3xGFP) internalize with the endocytic patches. Moreover, FRAP analysis in Fig.6 shows that Pak1 dynamics are distinct from that of the proxies of active Cdc42, Scd1 and Scd2. While Pak1 localization is active Cdc42 dependent, its dynamics are slower suggesting that once recruited Pak1 can exist at the membrane for at least a short period of time. A detailed analysis of Pak1 interaction with Cdc42 and its effectors will

explain Pak1 dynamics. We have revised our text to better reflect these ideas and explain our newly included FRAP analysis (p.19-20 line 361-371).

4. What are the assumptions in your model with regard to point 2 above? Your models currently have independent "degradation" terms (I would suggest that "detachment" would be a better term than "degradation" for this reaction) for Cdc42 and Pak1. This would seem to say that Pak1 can detach from Cdc42 and yet remain attached to the membrane until it is removed by endocytosis. Is that your view? Related to this, in your modeling time-series it would appear that Pak1 detaches from the membrane with a significant delay after Cdc42 has detached (e.g. Fig. 7A). If I read this correctly, this delayed detachment is responsible for the "offset" you analyze in Fig. 7. Would this disappear if the model coupled the detachment of Pak1 and Cdc42?

Response: Thank you for the great suggestion and questions. We agree that 'degradation' should be replaced by 'detachment'. We have made this change throughout the manuscript, particularly in model description that was inaccurate in the original manuscript.

Regarding the mechanism causing the 'offset' (i.e. a phase difference between cycles of Pak1 and Cdc42 shown in Fig 7A), our models do not impose any sequential molecular events such as the delayed detachment that the Editor mentioned. As an example, the dynamics of membrane-bound Pak1 at one tip is influenced by both activation by Cdc42 at the same location and detachment from the membrane simultaneously and continuously (at the macroscopic level, or concentration-wise, since our models do not describe behaviors of single molecules). Specifically, the dynamics is described as

$$\frac{dp}{dt} = f(c) - \delta p, \quad (1)$$

where p is membrane-bound Pak1 concentration at the tip, c is Cdc42-GTP concentration at the tip and δ is the detachment rate constant. f can be viewed as an activation Hill function. Eq 1 is simply an abbreviated form of the Eq 14 in our full model

description of the manuscript. Here, we first use Eq 1 to explain the Cdc42-Pak1 offset shown in Fig 7A under the assumption that sustained oscillation occurs. The ‘peak’ of Pak1 occurs when the right-hand side (RHS) $f(c) - \delta p = 0$, and it is just turning negative. This should happen sometime after c is at its peak in the same ‘cycle’, because when c is at its peak and starting to decrease (time point 1), $f(c)$ is still greater than δp (i.e. RHS of Eq 1 is positive). Therefore, at time point 1, p is still increasing. With continued decrease of c , and consequently the decreasing value of $f(c)$, RHS of Eq 1 becomes 0 (time point 2), after which p starts to decrease (Figure R1). As a rough analogy, we start to hit the brake pedal at time point 1, and the car will stop later at time point 2. This analogy only works for the ‘offset’ period between the two peaks because the p concentration will go down, but the car will not go backward by hitting the brake.

Figure R1. An illustration for the explanation of the phase difference between membrane-bound Cdc42-GTP and membrane-bound Pak1 concentrations.

The offset is crucial for the oscillatory dynamics characterized in this study. We therefore made additional perturbations to the dynamics of p in this revision. We found that either increasing the detachment rate constant δ or increasing δ and $f(c)$

proportionally reduced the peak offset (phase difference). The latter perturbation effectively made Pak1 a faster variable in general. With either perturbation, the oscillation dampened (Figure S4), suggesting the importance of relatively slow dynamics of Pak1 reflected in the offset.

This implicit-delay dependent oscillation is similar to the simplified model of the repressilator (https://biocircuits.github.io/chapters/09_repressilator.html). The three-variable model does not have any imposed delay or sequential molecular event. Yet, prominent phase differences were observed among the three variables.

5. Is independent detachment of Pak1 and Cdc42 critical for your conclusions? If so, it may be helpful to show some evidence for this key assumption. For example, FRAP analysis of Pak1 vs Scd1/Scd2 or Cdc42 could demonstrate that Pak1 is more stable (detaches more slowly). FRAP of Pak1 in control versus CK666 treatment could also provide important support for your assumption that Pak1 is less dynamic when endocytosis is inhibited.

Response: We thank you all for the suggestion to perform FRAP analysis of Pak1 dynamics. We performed FRAP on Pak1-mEGFP in DMSO and CK-666 treated conditions and find that the recovery of Pak1-mEGFP is significantly slower in CK-666 treated cells which we have now added as Fig.3F (p.16 lines 287-294). We also performed FRAP to compare Scd1-mNG, Scd2-GFP, and Pak1-mEGFP. We find that Pak1-mEGFP has slower dynamics than Scd1-mNG and Scd2-GFP. Unfortunately, the CRIB-GFP probe cannot be used for FRAP experiments and hence there is no way to accurately measure the dynamics of active Cdc42. Our data strongly suggests that Scd1 and Scd2 show similar oscillatory dynamics with active Cdc42. Hence, we compare the FRAP recovery rates of Pak1 with that of Scd1 and Scd2 (Fig.6D, p.19 lines 361-370). These additional results show further evidence that Pak1 does detach independent of Cdc42.

6. Does Pak1 regulate endocytosis globally or locally? You say (lines 349-350) that "Pak1 kinase has been shown to phosphorylate Myo1 *during endocytosis*" (emphasis mine) but you do not cite any paper for that and I am not aware of any evidence that phosphorylation occurs "in the act". Similarly, your model 2.1 assumes that local endocytosis rate is a Hill function of the local Pak1 concentration. But so far as I can tell, it could be that global Pak1 activity is required to maintain Myo1 phosphorylation for effective endocytosis. How important is it for your conclusions that Pak1 regulation occurs globally vs locally? If the model works equally well with global regulation, then perhaps you could change the model to be simpler. If, however, your conclusions are dependent on the steep local regulation currently assumed, then you might consider providing some evidence for local control.

Response: Thank you for pointing out these discrepancies. We do not intend to claim that Pak1 phosphorylates Myo1 in the act "during" endocytosis and have adjusted our text accordingly. We now added that Pak1 phosphorylates Myo1 "for" endocytosis (Attanapola et al., 2009)(p.21 line 407, p.23 line 445-446). Globally, loss of Pak1 function results in loss of polarity of the cell and, in line with this, endocytosis is no longer properly polarized. Locally, we see that *pak1-ts* mutants show hindered endocytic dynamics as we see less internalization and shorter patch lifetimes as well as more stalled and failed events in *pak1-ts* mutants when compared to *pak1+* as shown in Fig.9. We are currently investigating the role of Pak1 in regulating endocytosis.

In terms of the mathematical model, the inclusion of activation of endocytosis by Pak1 locally does not change the general behaviors of the model significantly. In other words, regardless of whether Pak1 promotes endocytosis locally (i.e. whether the Hill function in Eq. 18 mentioned by the Editor is included in Model 2 or not), the model shows bipolarity and anti-correlated oscillations. In the generic model where the negative regulator X is generic, we did not include the Pak1-endocytosis regulation, whereas in the specific model where X is replaced by Pak1, we included the regulation. The latter was inspired by experimental data mentioned above. Nonetheless, we did observe minor difference between the two versions (with or without Pak1-endocytosis regulation) of Model 2, e.g.

the inclusion of the regulation resulted in larger amplitude of Pak1 oscillation, but the amplitudes for Cdc42-GTP and Scd1 were smaller. We have now clarified these points in supplementary materials of the models and Fig.S4C. We plan to investigate the precise roles of Pak1-endocytosis regulation with more modeling and experimental work in the future.

Reviewer #1,

Endocytosis is known to modulate cell growth and shape by affecting cell polarization and exocytosis, but the underlying mechanisms are not clearly understood. In the rod-shaped fission yeast, the Cdc42-GTP (indicated by CRIB-3xGFP) displays anti-correlated oscillations between the cell ends, which is essential for bipolar growth in this organism. This oscillatory behavior requires a positive feedback mechanism involving Scd1 (GEF for Cdc42) and Scd2 (scaffold) and a time-delayed negative feedback mechanism involving Pak1 (an effector of Cdc42). In this study, the authors report that disruption of endocytosis by the Arp2/3-specific inhibitor CK-666 disrupts CRIB oscillation and disables the positive feedback mechanism (by decreasing the localization of Scd1 and Scd2 at the cell ends). This is an important finding that is well supported by the data. Through further experimentation and modeling, the authors conclude that the Cdc42-dependent recruitment of Pak1 to a cell end, when reached to a critical level, will disable the positive feedback mechanism presumably phosphorylating Scd1 and Scd2, causing Scd1-Scd2 dissociation (as suggested by previous studies), and that endocytic removal of Pak1 is the key for the return of Cdc42 activation to the cell end. Furthermore, the authors conclude that Pak1 may stimulate its own removal by facilitating endocytosis. While this Pak1-centered idea is quite attractive, further experiments and clarification are required to establish its validity.

Major points:

1. To test the role of endocytic removal of Pak1 in Cdc42 oscillations as well as the role of Pak1 in endocytosis, the same of panel of oscillation and endocytosis makers (i.e., CRIB, Scd1, and Fim1) should be examined by time-lapse analysis in cells [such as GFP-pak1-as (analog sensitive) cells] where Pak1 localization can be monitored, and its activity can be acutely inactivated. These data should be compared to the data derived from CK-666-treated cells to determine their similarities and differences. This will help determine to what degree the impact of endocytosis on Cdc42 oscillation is mediated by Pak1 activity, to what degree Pak1 controls endocytosis, and to what degree Pak1 is removed by endocytosis. If the pak1-as is not available, the same experiments should be performed using the pak1-Ts strain (not simply monitoring the Scd1 intensity as done in Fig. 4B).

Response: We do appreciate the suggestion to parse apart the degrees to which Pak1 controls endocytosis and/or is removed via endocytosis. The elegant experiments that you have proposed would be a great way to test this. However, we believe that this extensive analysis is beyond the scope of this current work. At present, we are reporting that Pak1 is removed during endocytosis and this suggestion would help delve further into the details of that mechanism and will be more suited for future work.

2. CK-666 treatment appears to mainly dampen the dynamics of CRIB (Fig. 1A and Fig. S1B, right) and Pak1 (Fig. 3C, right) at one end, likely the old end (Fig. S1B, right), while the oscillations (or on/off) of CRIB and Pak1 at the other end appear normal. Why and how?

Response: We find that initially after CK666 treatment, the cells appear to show asymmetric Cdc42 activation at one end while it is almost lost at the other end. With prolonged treatment (35min+) both ends show significant loss of CRIB-3xGFP signal. Moreover, the signal even at the initial active end is dampened in these cells. We have revised the representative image in Fig.1A to better reflect our data. We also added a

supplemental movie to show how CRIB-3xGFP signal at the ends changes during CK-666 treatment. The protein dynamics are altered at the end that still oscillates. In our Pak1-mEGFP graphs (revised Fig.4C), we show that the duration of oscillations can be longer and thereby the wavelength and periodicity can be longer and less consistent when compared to controls. In the case of CRIB-3xGFP oscillations (Revised Fig. 1C) the peak of intensity is significantly lower. We have also now included FRAP data that further show that the half-life of fluorescence recovery at the active end is slower, which would coincide with longer periods/wavelengths. It is likely that there is a limit to how drastically different the dynamics can be as the inherent properties of these protein interactions and functions are not being changed. Further, our models also show that extreme perturbations, which may not be biologically possible, would break the system and we would then see no oscillations at either end (Fig. S4) (p.19 line 347-360).

3. The impact of Myo1 deletion on Scd1 and Pak1 dynamics (Fig. 5) seems to be stronger than that caused by CK-666 treatment, yet the effect of Myo1 deletion on CRIB dynamics appears to be weaker than that by CK-666 treatment. Why?

Response: While the *myo1Δ* and CK-666 treatment both show endocytic defects, it is important to note that *myo1Δ* cells are viable and growing while CK-666 treated cells stop growth. Thus, it is expected that Cdc42 activity will not be as dramatically impacted in *myo1Δ* as it is in CK-666 treated cells. As for the Scd1 levels, we see 2.5-fold decrease in Ck-666 treated cells while we observe a 1.2-fold decrease in *myo1Δ* cells. This is in agreement with the fact that active Cdc42 is not as dampened in the *myo1Δ* cells. As for the Pak1 dynamics, this can be regulated by not just Cdc42 but also protein availability due to turnover. It is likely that protein synthesis and degradation rate are different in the viable *myo1Δ* strain compared to the non-growing CK666 treated cells. These differences may contribute to the significant changes in Pak1 dynamics in *myo1Δ* cells.

Minor points:

1. *Fig. 1, A and B: while anti-correlated oscillations are maintained in for3 Δ cells, the CRIBX3GFP signal at the cell ends appear much stronger than that in WT cells. If so, how?*

Response: Regarding the minor points, we have also noticed the robust oscillations and Cdc42 activity in *for3 Δ* mutants. We find this phenomenon interesting, and will be a separate study from this current work.

2. *Line 235, change "Fig. 3B" to "Fig. 2B"*

3. *Line 301, Change "Fig. 5A, B" to "Fig. 4A, B"*

Response: Thank you for highlighting some of the inconsistencies with our figure numbering. They have been corrected.

4. *Discuss more about the similarities and differences between the Scd1-Scd2-pak1 in fission yeast and the Cdc24-Bem1-Cla4 in budding yeast in terms of their roles in controlling the dynamics of Cdc42 polarization. The paper on the role of Cla4 in regulating polarized growth by phosphorylating Cdc24 (Gulli et al., Mol. Cell. 2000 by Mathias Peter's group) and the paper on the dispersal of Cdc42 by endocytosis (Irazoqui et al., MBoC. 2005 by Daniel Lew's group) should be included for discussion.*

Response: We apologize for this oversight and have added these references to the manuscript.

Reviewer #2,

This study by Harrell et al focuses on a potential mechanism of removal of Pak1 kinase-mediated negative feedback, which is essential for Cdc42 oscillations between two cell ends that are responsible for polarized bipolar growth of fission yeast S. pombe cells. The authors discovered that blocking endocytosis through treatment with Arp2/3 complex inhibitor CK-666 or the use of myo1 delta mutant (which is viable but has severe

endocytosis defect) stops Cdc42 and Pak1 oscillations, decreases accumulation of Cdc42 GEF Scd1, and increases accumulation of Pak1. Through mathematical modeling and excellent 2-color live cell imaging the authors provide strong evidence that Pak1 is responsible for delayed negative feedback, which is essential for Cdc42 oscillations. For this model to work, Pak1 needs to be eventually removed and, based on their CK-666 and myo1delta data the authors propose that Arp2/3 complex-mediated branched actin-dependent endocytosis is directly responsible for Pak1 removal. Indeed, when the authors looked at Pak1 at endocytic sites, they observed that Pak1 briefly disappeared from the membrane at the moment an endocytic actin patch internalized. However, the authors did not see Pak1 internalizing with the patch, which is very surprising and casts doubt on the authors' overarching conclusion that endocytosis is directly responsible for Pak1 removal. This is a major concern and below I propose additional experimentation that may verify or defy this conclusion.

Overall, the data in this manuscript are of good quality but I wish I could say the same about the writing and especially citations, as detailed below. The manuscript in its current form is not acceptable for publication in JCB and needs major revision as well as a little bit of additional experimentation.

Major concerns:

(1) The observations that blocking endocytosis with CK-666 or myo1delta results in increased accumulation of Pak1 is very convincing and extremely intriguing. It does suggest that actin-mediated endocytosis does play some important role in Pak1 removal. The authors' observation that Pak1 level on the membrane at the endocytic site drop at the exact moment of actin patch internalization is equally convincing but the fact that the authors do not see Pak1 internalizing with the patch suggests that endocytosis does not remove Pak1 directly (i.e. removing Pak1 that is associated with the membrane of an endocytic vesicle) but does so indirectly through some thus far unknown

mechanism. The authors can address this concern both experimentally and by changing the wording throughout the ms:

(a) Would it make sense to go back and take a look with a very sensitive detector to make absolutely sure that no Pak1 is internalizing with the patch? In doing so, the authors would need to be very careful to include Fim1-mCherry only control for bleed through.

Response: At this time, it is beyond the scope of this work to be able to fully elucidate the mechanism or fate of Pak1 that are removed by endocytosis or an indirect mechanism. We have currently revised Figure 8A to show punctate localization of Pak1 and its colocalization with Fim1 prior to internalization. Please see the response to the Editor above for further details (p.1-2).

Currently, technical limitations prevent us from investigating if there is any low levels of Pak1 on internalized patches. We have recently attempted to observe the colocalization of Pak1 on the internalized endocytic patch using a Zeiss Airyscan microscope and still do not see colocalization after internalization.

(b) The authors have an excellent opportunity to test their conclusions by repeating Pak1-EGFP and Fim1-mCherry 2-color imaging in myo1delta cells. Over half of actin patches in myo1delta fail to internalize. So, if the authors are right, they should see that in myo1delta cells Pak1 does not disappear from the membrane when patches do not internalize. If Pak1 continues to disappear in non-internalizing patches, that would suggest that some other mechanisms rather than direct endocytosis are at play.

Response: We have found that, while the *myo1Δ* strains are viable, they are extremely sensitive to tagging of polarity proteins. So far, we have not successfully made a *myo1Δ* Pak1-mEGFP Fim1-mCherry strain. The strains are able to mate and form diploids but

the spores generated do not show *myo1* Δ with both fluorophores. We had a similar struggle in making a *myo1* Δ Pak1-mEGFP Cdc42-mCherry strain, suggesting that these combinations are synthetically lethal.

(c) Throughout the ms, including the title, the authors may want to consider softening the language by replacing statements like "endocytosis removes Pak1" with something less direct like "endocytosis plays a role in, or is important for Pak1 removal". Likewise, it would be better to re-word the title into something like "Arp2/3- and Myo1-dependent removal of Pak1 facilitates..." Also, please consider softening the title of Figure 6 "Pak1 is removed from the cell end via endocytosis".

Response: We agree that our wording may be too definitive and have worked to soften the language accordingly. We have now changed our title to "Arp2/3-dependent endocytosis allows Cdc42 oscillations by removal of Pak1-mediated negative feedback". Similarly worded headings for sections and figures 8 and 9 have now been changed.

(2) There are serious problems with citations throughout the manuscript. The authors, like this reviewer, should invest some time into double- and triple-checking their references to make sure that the references appropriately support their statements. Here are just five (out of many) examples:

(a) Introduction, p7: Active Cdc42 also triggers branched actin polymerization, necessary for endocytosis in yeasts, through the WASP protein, Wsp1, and the type 1 myosin, Myo1 (Bendezú and Martin, 2011; Gachet and Hyams, 2005; Kolluri et al., 1996; Kovar et al., 2011; Landino et al., 2021; Lechler et al., 2000; Lee et al., 2000; Sirotkin et al., 2005).

First of all, the statement that "Active Cdc42 also triggers branched actin polymerization ... through the WASP protein, Wsp1..." is factually not true for yeast. Budding yeast WASp Las17 lacks GTPase binding domain (GBD) and it is not clear if S. pombe Wsp1

has a GBD and as far as know, nobody looked at it in S. pombe experimentally. Even in mammalian cells, while it is true that WASp and N-WASp are activated by Cdc42, it is still not clear if this mode of activation occurs at the sites of endocytosis.

Now to the references. Bendezú and Martin, 2011, Gachet and Hyams, 2005, Landino et al., 2021, Lee et al., 2000, Sirotkin et al., 2005 make no mention of Cdc42 activating Wsp1. Kovar et al., 2011 only mention Cdc42 in the context of actin cables. Lechler et al., 2000 (as well as Lechler et al., 2001, which authors failed to cite) is a budding yeast work that talks about Cdc42-Pak1-myosin 1 activation but not Wsp1. Kolluri et al. is a mammalian WASp reference. Thus, none of the papers cited by the authors support their statement.

(b) Discussion, p29: Likewise, Cdc42 activation also promotes actin polymerization for endocytosis and exocytosis (Adamo et al., 2001; Campbell et al., 2022; Estravis et al., 2012; Estravis et al., 2017; Estravis et al., 2011; Gachet and Hyams, 2005; Kovar et al., 2011; Lamas et al., 2020a; Sagot et al., 2002).

Every single one of these 9 references deals with the role of Cdc42 in exocytosis only. No references are provided for the role of Cdc42 in endocytosis.

(c) Discussion, p26: Bipolarity is established when the new end is able to overcome the old end dominance and robustly activate Cdc42 (Mitchison and Nurse, 1985).

Unfortunately, this classic paper makes no mention of Cdc42.

(d) Results, p21: While the role for Pak1 kinase in endocytosis has not been fully investigated, it has been shown to phosphorylate Myo1 during endocytosis (Attanapola et al., 2009; Lee et al., 2000).

Lee et al., 2000 have not shown that. They only mentioned a potential role of Pak1 in phosphorylating Myo1 in the Discussion, based on prior budding yeast work, which authors of the current ms neglected to cite.

(e) Discussion, p24: Arp2/3 dependent endocytosis is known to be required for growth in S. pombe and other fungal species as well as mammalian cells with high membrane tension (Aghamohammadzadeh and Ayscough, 2009; Basu et al., 2014b; Epp et al., 2010).

All three of these are yeast references. The authors must cite Boulant et al (2011) Nature Cell Biology for mammalian cells.

Response: We apologize for this oversight as we encountered numerous technical issues when working on our citations. We have diligently rechecked and corrected all the references in preparation for publication.

(3) The ms is rather on the verbose side and can greatly benefit from some serious cutting and editing. This is especially evident in the Introduction. Additionally, several sections of the Discussion are redundant with the Introduction, for example, the entire first paragraph of page 26.

Response: We have actively tried to remove as much clutter as possible. We write with the intention of making the manuscript clear to those who are not directly in our field and try to present and reiterate the relevant information accordingly. But we do take heed to your suggestion and have removed excess additional text in the revised manuscript. We have also removed several areas of text to eliminate some redundancy.

Minor concerns:

Results, p21. Unfortunately, the authors switched from correctly calling Fim1-mCherry patches endocytic actin patches (or simply actin patches or patches) to calling them endocytic vesicles. This is unfortunate because they made this switch while describing endocytic defects in pak1-ts mutant, where actin patches marked by Fim1-mCherry still form but the endocytic vesicles may fail to form, invaginate or internalize. I suggest the authors change the wording on this page from "endocytic vesicles" to "actin patches".

Response: We agree that the terminology here was inconsistent. We have now changed the wording to refer to them as patches throughout the manuscript.

The authors may want to consider moving their excellent Fig. 7 to earlier in the paper, for example right after Fig. 2, as the data in current Fig. 7 strongly support the role of Pak1 as being responsible for the delayed negative feedback.

Response: Thank you for the suggestions. We agree that moving the phase-shift figure to be earlier in the manuscript has enhanced the flow of the paper.

Introduction, p7: The statement that "While Cdc42 has been shown to regulate endocytosis and actin organization" needs to be supported by appropriate references if such references exist, especially as it pertains to endocytosis.

Response: We have revised this to indicate that Cdc42 has been shown to regulate actin organization and that we hypothesize that Pak1 may regulate endocytosis, which ultimately promotes Pak1 removal. Future work from our lab will investigate this further. Please see our response to the Editor above for further details (p.2).

Discussion, p27: "... removal of negative feedback by Pak1-dependent endocytosis..." Do you mean "removal of Pak1-dependent negative feedback by endocytosis"?

Response: We did mean “Pak1-dependent endocytosis” to signal to the readers that Pak1 plays a role in facilitating endocytosis that may ultimately be promoting its own removal. For clarity we have changed the statement to say, “disruption of negative feedback via endocytosis-dependent Pak1 removal” (p.27 line 521).

Figure 2B. Would it make sense to add labels Model 2, Cytoplasm, Cell Ends directly to this panel rather than rely on the Figure Legend alone?

Response: Yes, we added the labels directly to the panel in the new version.

Figure 3B vs 3D. The Pak1-mEGFP signal in the CK-666 treated image in Fig. 3B looks dimmer than in DMSO control image, contrary to quantitation in Fig. 3D. Were the two panels in Fig. 3B equally adjusted or is there an error?

Response: We apologize for any confusion that we have caused. We have revised our representative images of Pak1-mEGFP in DMSO and CK-666 conditions. In this case, the cytoplasm of DMSO treated cells is darker than that of the CK-666 treated cells. The lower cytoplasmic signal in Ck-666 treated cells are due to the increased protein levels at the membrane.

Figure 4C. Would it make sense to add an asterisk or some other label to the green negative error to highlight that this is a new addition unique to Model 3?

Response: Thank you for this suggestion. We have added an asterisk to the green arrow (Fig 5D)

There are many typos or missing spaces scattered throughout the ms.

Response: We apologize for the errors throughout the manuscript and have diligently corrected these in the revised version.

Reviewer #3,

This manuscript by Harrell, Liu et al., uncovers a regulatory link between clathrin-mediated endocytosis and the oscillations of active Cdc42 at sites of polarized growth in fission yeast. This is a very interesting paper and has important implications in understanding the coupling of membrane trafficking and polar growth. These mechanisms are likely widely conserved in many branches of eukaryotic life.

The experiments are carefully done and clearly presented. The conclusions are well supported by the provided genetic and cell biological data. I cannot comment on the technical aspects of the mathematical models, but they are well connected to the biological experiments and give a strong quantitative framework for thinking about the different possible regulatory networks. The manuscript is clearly written and logically structured. I believe it would be of great interest for the readership of JCB.

I have a few suggestions for further experiments. These are not experiments that I consider essential for the current manuscript, but they might add value to the work.

*The exact mechanism by which Arp2/3 complex has an effect on Pak1 remains open. Is the effect mediated directly by the assembly of actin patches or does it require the formation of endocytic vesicles? The experiments with myosin mutants could shed some light on this question, but the authors do not describe in detail what is the phenotypic effect of the myo1 deletion on endocytic progression. Has this been described in previous studies in *S. pombe*? Knowing the effect of myosin deletion on endocytosis could help to understand the mechanism by which Arp2/3 activity regulates Pak1. The authors could do similar analyses on the myo1 mutant as they did in fig6 to characterize the pak1-ts phenotype.*

Response: We agree that we cannot yet distinguish if the loss of Pak1 via endocytosis is a direct or indirect mechanism. We thank the reviewer for recognizing our work as interesting and relevant, but mechanistic details will have to be reserved for future studies. For now, we have attempted to clarify our findings in the text and with updates to Fig. 8. We show that Pak1-mEGFP exists as distinct puncta that colocalizes with Fim1-mCherry patches prior to patch internalization. Please see our response to the Editor above for further details (p.2)

Another endocytic mutant that could be informative in this context is the deletion of end4. This mutant blocks endocytosis, but allows robust Arp2/3 activation and actin assembly. However, the strong endocytosis block in the end4 mutants might make the experiments difficult.

Response: As you mention, the strong endocytic defects of many endocytic mutants, including *end4*, make experimentation difficult. *end4* mutants are also depolarized and the lack of polarity would hamper our ability to do meaningful analysis of oscillations.

Textual points:

- To me the title of the manuscript seems to imply that the authors are showing a direct connection between the Arp2/3 complex and Pak1, which obviously is not the case. I would rethink the title so that it better reflects the conclusions of the paper. For example: "Arp2/3 complex -dependent endocytosis periodically removes....".

Response: We also agree that the current title doesn't accurately reflect what we are showing with our work. It is now "Arp2/3-dependent endocytosis allows Cdc42 oscillations by removal of Pak1-mediated negative feedback"

- *The authors state that Cdc42 regulates polarized growth in most eukaryotes. (lines 17, 48, 64). Maybe this is common knowledge in the polarization field, but could the authors cite the work that demonstrates this.*

Response: We have added citations to indicate the role of Cdc42 in regulating polarized growth.

- *On line 360 the authors cite Nakano et al., 2001. I don't think that this paper showed that "fimbrin internalizes into the cytoplasm with the endocytic vesicle".*

Response: Thank you for bringing this to our attention, we have noticed several issues with the citation software, and we have diligently corrected all the citations in the revised manuscript.

- *In the discussion section I was missing some discussion about the mechanism by which endocytosis removes Pak1. Uncovering this mechanisms is clearly beyond the scope of this manuscript, but it would be good to highlight this as an important open question and maybe propose some possibilities. There was actually some discussion about this point in the results section. Maybe that text would work better in the discussion section?*

Response: We have now added additional discussion regarding the mechanism by which endocytosis removes Pak1. In the discussion we have mentioned that "The Cdc42 GEF Gef1 localizes to the endocytic patches in a F-BAR protein Cdc15-dependent manner (Hercyk and Das, 2019). Cdc15 has been shown to bind Myo1 at these patches (Arasada and Pollard, 2011; Arasada et al., 2018; Carnahan and Gould, 2003). Thus, it is possible that Pak1 is part of a protein complex including its substrate Myo1, Cdc42, Gef1 and Cdc15. Both Cdc15 and Myo1 disappear from the membrane upon patch internalization (Arasada and Pollard, 2011). It is possible that Pak1 is lost along with that complex." (p.26, lines 490-497)

May 22, 2024

RE: JCB Manuscript #202311139R

Dr. Maitreyi Das
Boston College
Biology
140 Commonwealth Avenue
Chestnut Hill, MA 02467

Dear Dr. Das,

Thank you for submitting the revised version of your manuscript. We believe that the revisions have significantly improved the paper, particularly the addition of FRAP data as well as the reinterpretation of Pak1 localization. Having assessed these and the other major points requested by reviewers, we have determined that their significant concerns are addressed in this revised work. Therefore we would be happy to publish your paper in JCB pending minor revisions and formatting changes necessary to meet journal guidelines (see details below). In particular:

I did not understand how to infer off-rates from a half-end FRAP: how do you know whether recovery is due to exchange with the cytoplasm vs diffusion from the unbleached half-end? A few example FRAP-series images would allow one to see whether recovery of the bleached part is accompanied by dimming of the unbleached part (as expected for diffusion-based recovery) or not (as expected for on/off recovery). If diffusion seems important, then you would need a way to infer on/off rates in your comparisons between different probes.

The new deconvolved image showing Pak1 colocalization with Fim1 puncta is striking. But one image is not sufficient to make a convincing point, and the kymograph does not seem to show similar puncta. Perhaps a series of images (preferably including one of the other imaging modalities you mentioned in your rebuttal, which would be useful to alleviate worries about deconvolution artifacts) would make this more persuasive.

I noticed some remaining issues with referencing, and had some clarification questions:

Line 96-98: Cla4 phosphorylates Cdc24 "to prevent its interaction with Bem1" (Gulli; Kuo; Rapali). But Kuo specifically says that the phosphorylation inhibits GEF activity WITHOUT affecting interaction with Bem1 (as did Bose et al 2001), and Rapali showed reduced GEF activity as well. It would be more accurate to say "to reduce its GEF activity".

Line 111-112: Myo1 regulates endocytosis "by binding Arp2/3...". That is ONE way for Myo1 to regulate endocytosis, but Drubin and Kaksonen have shown that in addition to that the type I myosins are important for membrane anchoring and force production?

Line 236-237: "membrane bound Scd1 inhibitors have been found experimentally (Butty et al 2002)". I am not aware of any data in that paper that would support your assertion. Did you mean a different paper?

Line 243: "a representative oscillation-promoting parameter set". What makes a parameter set "representative"? If it is arbitrary, then checking your results in a couple of other (not near neighbor) oscillation-promoting parameter sets would seem worthwhile.

Line 267-269: Pak1 is "part of a time-delayed negative feedback ...(Marcus; Otilie; Tu)". But these three papers preceded the recognition of negative feedback. To make this statement you need more recent references.

Line 285-287: You cite a modeling Figure (S2A) to support an experimental statement, which seems inappropriate.

A. MANUSCRIPT ORGANIZATION AND FORMATTING:

Full guidelines are available on our Instructions for Authors page, <http://jcb.rupress.org/submission-guidelines#revised>. Submission of a paper that does not conform to JCB guidelines will delay the acceptance of your manuscript.

1) Text limits: Character count for Articles is < 40,000, not including spaces. Count includes abstract, introduction, results, discussion, and acknowledgments. Count does not include title page, figure legends, materials and methods, references, tables, or supplemental legends.

2) Figures limits: Articles may have up to 10 main figures and 5 supplemental figures/tables.

3) Figure formatting: Scale bars must be present on all microscopy images, including inset magnifications. Molecular weight or nucleic acid size markers must be included on all gel electrophoresis. Please avoid pairing red and green for images and graphs to ensure legibility for color-blind readers. If red and green are paired for images, please ensure that the particular red and green hues used in micrographs are distinctive with any of the colorblind types. If not, please modify colors accordingly or provide separate images of the individual channels.

** Please include a scale bar on Fig 1A and 8A (all panels).

4) Statistical analysis: Error bars on graphic representations of numerical data must be clearly described in the figure legend. The number of independent data points (n) represented in a graph must be indicated in the legend. Statistical methods should be explained in full in the materials and methods. For figures presenting pooled data the statistical measure should be defined in the figure legends. Please also be sure to indicate the statistical tests used in each of your experiments (either in the figure legend itself or in a separate methods section) as well as the parameters of the test (for example, if you ran a t-test, please indicate if it was one- or two-sided, etc.). Also, if you used parametric tests, please indicate if the data distribution was tested for normality (and if so, how). If not, you must state something to the effect that "Data distribution was assumed to be normal but this was not formally tested."

** Please indicate n in the figure legends for Figure 1D, 4G, 5B, 6D, 6F, 7B, 8D, 9C/D.

5) Abstract and title: The abstract should be no longer than 160 words and should communicate the significance of the paper for a general audience. The title should be less than 100 characters including spaces. Make the title concise but accessible to a general readership.

** For clarity and to emphasize the central findings here, you may wish to alter your title to:

Arp2/3-dependent endocytosis ensures Cdc42 oscillations by removal of Pak1-mediated negative feedback

6) Materials and methods: Should be comprehensive and not simply reference a previous publication for details on how an experiment was performed. Please provide full descriptions in the text for readers who may not have access to referenced manuscripts. We also provide a report from SciScore and an associate score, which we encourage you to use as a means of evaluating and improving the methods section.

** Please rephrase "Standard fluorescence intensity method" as this phrase is unclear. Please also describe FRAP analysis, as the current methods indicate a bleaching exposure of 150 ms at 5% power yet the acquisition exposure is 250 ms at 15% power.

7) Please be sure to provide the sequences for all of your primers/oligos and RNAi constructs in the materials and methods. You must also indicate in the methods the source, species, and catalog numbers (where appropriate) for all of your antibodies. Please also indicate the acquisition and quantification methods for immunoblotting/western blots.

8) Microscope image acquisition: The following information must be provided about the acquisition and processing of images:

a. Make and model of microscope

b. Type, magnification, and numerical aperture of the objective lenses

c. Temperature

d. Imaging medium

e. Fluorochromes

f. Camera make and model

g. Acquisition software

h. Any software used for image processing subsequent to data acquisition. Please include details and types of operations involved (e.g., type of deconvolution, 3D reconstitutions, surface or volume rendering, gamma adjustments, etc.).

** At least four microscopes are mentioned. Please indicate which data were acquired with which instrument in the methods.

10) Supplemental materials: There are strict limits on the allowable amount of supplemental data. Articles may have up to 5 supplemental figures. Please also note that tables, like figures, should be provided as individual, editable files. A summary of all supplemental material should appear at the end of the Materials and methods section.

** Please include greater details in legends for supplementary videos.

13) ORCID IDs: ORCID IDs are unique identifiers allowing researchers to create a record of their various scholarly contributions in a single place. At resubmission of your final files, please provide an ORCID ID for all authors.

15) A data availability statement is required for all research article submissions. The statement should address all data underlying the research presented in the manuscript. Please visit the JCB instructions for authors for guidelines and examples of statements at (<https://rupress.org/jcb/pages/editorial-policies#data-availability-statement>).

The source code for all custom computational methods published in JCB must be made freely available as supplemental material hosted at www.jcb.org. Please contact the JCB Editorial Office to find out how to submit your custom macros, code for custom algorithms, etc. Generally, these are provided as raw code in a .txt file or as other file types in a .zip file. Please also include a one-sentence summary of each file in the Online Supplemental Material paragraph of your manuscript.

Journal of Cell Biology now requires a data availability statement for all research article submissions. These statements will be published in the article directly above the Acknowledgments. The statement should address all data underlying the research presented in the manuscript. Please visit the JCB instructions for authors for guidelines and examples of statements at (<https://rupress.org/jcb/pages/editorial-policies#data-availability-statement>).

B. FINAL FILES:

Thank you for your attention to these final processing requirements. Please revise and format the manuscript and upload materials within 7 days. If you need an extension for whatever reason, please let us know and we can work with you to determine a suitable revision period.

Thank you for this interesting contribution, we look forward to publishing your paper in Journal of Cell Biology.

Sincerely,

Daniel Lew
Monitoring Editor
Journal of Cell Biology

Tim Fessenden
Scientific Editor
Journal of Cell Biology

Dear Dr. Lew

Thank you for email and for accepting to publish the manuscript. We have made the suggested revisions in the updated manuscript. The specific responses to the comments are mentioned below.

I did not understand how to infer off-rates from a half-end FRAP: how do you know whether recovery is due to exchange with the cytoplasm vs diffusion from the unbleached half-end? A few example FRAP-series images would allow one to see whether recovery of the bleached part is accompanied by dimming of the unbleached part (as expected for diffusion-based recovery) or not (as expected for on/off recovery). If diffusion seems important, then you would need a way to infer on/off rates in your comparisons between different probes.

We now provide images of the cells displaying recovery after photobleaching in the supplement (Supplementary Figure S3C and S5B). We find that upon bleaching one half of the cell end, the fluorescent signal at the non-bleached half also decreases, but this end rapidly regains the signal even as the bleached half recovers. This indicates that the signal recovery at the bleached half of the cell end is not due to lateral diffusion from the nonbleached region but rather from protein uptake from the cytoplasm. We have also mentioned this in the text of the paper (lines 298-301, 374-378). For your reference, we have provided representative signal recovery data from two cells for Scd1-mNG, Scd2-GFP and Pak1-mEGFP below.

The new deconvolved image showing Pak1 colocalization with Fim1 puncta is striking. But one image is not sufficient to make a convincing point, and the kymograph does not seem to show similar puncta. Perhaps a series of images (preferably including one of the other imaging modalities you mentioned in your rebuttal, which would be useful to alleviate worries about deconvolution artifacts) would make this more persuasive.

We now provide additional images of Pak1-mEGFP and Fim1-mCherry in Figure 8B. In addition to the deconvolution image in figure 8A we now show images taken with a Zeiss Airyscan super resolution microscopy. Here too we see regions of high intensity Pak1-mEGFP at the cell ends that appear as patches. In addition, here we can clearly see Pak1-mEGFP patches along cell sides that also overlap with Fim1-mCherry.

I noticed some remaining issues with referencing, and had some clarification questions:

Line 96-98: Cla4 phosphorylates Cdc24 "to prevent its interaction with Bem1" (Gulli; Kuo; Rapali). But Kuo specifically says that the phosphorylation inhibits GEF activity WITHOUT affecting interaction with Bem1 (as did Bose et al 2001), and Rapali showed reduced GEF activity as well. It would be more accurate to say "to reduce its GEF activity".

We have now made the suggested change in the text.

Line 111-112: Myo1 regulates endocytosis "by binding Arp2/3...". That is ONE way for Myo1 to regulate endocytosis, but Drubin and Kaksonen have shown that in addition to that the type I myosins are important for membrane anchoring and force production?

We have added these references and also mentioned Myo1's role in membrane anchoring and force production in the text.

Line 236-237: "membrane bound Scd1 inhibitors have been found experimentally (Butty et al 2002)". I am not aware of any data in that paper that would support your assertion. Did you mean a different paper?

We have corrected the reference and the statement now reads "Furthermore, the PAK kinase has been found to inhibit Scd1/Cdc24 experimentally (Das et al., 2012; Gulli et al., 2000; Kuo et al., 2014; Rapali et al., 2017)."

Line 243: "a representative oscillation-promoting parameter set". What makes a parameter set "representative"? If it is arbitrary, then checking your results in a couple of other (not near neighbor) oscillation-promoting parameter sets would seem worthwhile.

We now show oscillations with additional parameter sets in Supplementary figure S2C,D and Table S2.

Line 267-269: Pak1 is "part of a time-delayed negative feedback ...(Marcus; Otilie; Tu)". But these three papers preceded the recognition of negative feedback. To make this statement you need more recent references.

We now provide the following references for this statement, Das et al., 2012; Das and Verde, 2013; Kuo et al., 2014.

Line 285-287: You cite a modeling Figure (S2A) to support an experimental statement, which seems inappropriate.

We have removed this citation.